# Semantic segmentation of sparse irregular point clouds for leaf/wood discrimination

**Yuchen Bai**[1]     **Jean-Baptiste Durand**[2][*]     **Grégoire Vincent**[2]     **Florence Forbes**[1]

[1]Univ. Grenoble Alpes, CNRS, Inria, Grenoble INP, LJK, Grenoble, France
[2]AMAP, Univ. Montpellier, CIRAD, CNRS, INRAE, IRD, Montpellier, France
{yuchen.bai,florence.forbes}@inria.fr
jean-baptiste.durand@cirad.fr
gregoire.vincent@ird.fr

## Abstract

LiDAR (Light Detection And Ranging) has become an essential part of the remote sensing toolbox used for biosphere monitoring. In particular, LiDAR provides the opportunity to map forest leaf area with unprecedented accuracy, while leaf area has remained an important source of uncertainty affecting models of gas exchanges between the vegetation and the atmosphere. Unmanned Aerial Vehicles (UAV) are easy to mobilize and therefore allow frequent revisits, so as to track the response of vegetation to climate change. However, miniature sensors embarked on UAVs usually provide point clouds of limited density, which are further affected by a strong decrease in density from top to bottom of the canopy due to progressively stronger occlusion. In such a context, discriminating leaf points from wood points presents a significant challenge due in particular to strong class imbalance and spatially irregular sampling intensity. Here we introduce a neural network model based on the Pointnet ++ architecture which makes use of point geometry only (excluding any spectral information). To cope with local data sparsity, we propose an innovative sampling scheme which strives to preserve local important geometric information. We also propose a loss function adapted to the severe class imbalance. We show that our model outperforms state-of-the-art alternatives on UAV point clouds. We discuss future possible improvements, particularly regarding much denser point clouds acquired from below the canopy.

## 1   Introduction

In the past decades, LiDAR technology has been frequently used to acquire massive 3D data in the field of forest inventory (Vincent et al. [1]; Ullrich & Pfennigbauer [2]). The acquisition of point cloud data by employing LiDAR technology is referred to as laser scanning. The collected point cloud data provides rich details on canopy structure, allowing us to calculate a key variable, leaf area, which controls water efflux and carbon influx. Monitoring leaf area should help in better understanding processes underlying flux seasonality in tropical forests, and is expected to enhance the precision of climate models for predicting the effects of global warming. There are various types of vehicles for data collection, with ground-based equipment and aircraft being the most commonly employed. The former operates a bottom-up scanning called terrestrial laser scanning (TLS), providing highly detailed and accurate 3D data. Scans are often acquired in a grid pattern every 10 m and co-registered into a single point cloud. However, TLS requires human intervention within the forest, which is laborious and limits its extensive implementation. Conversely, airborne laser scanning (ALS) is much faster and can cover much larger areas. Nonetheless, the achieved point density is typically two

---

[*]Corresponding author.

37th Conference on Neural Information Processing Systems (NeurIPS 2023).

orders of magnitude smaller due to the combined effect of high flight altitude and fast movement of the sensor. Additionally, occlusions caused by the upper tree canopy make it more difficult to observe the understory vegetation.

In recent years, the development of drone technology and the decreasing cost have led to UAV laser scanning (ULS) becoming one favored option (Brede et al. [3]). It does not require in-situ intervention and each flight can be programmed to cover a few hectares. The acquired data is much denser than ALS (see Figure 1(a) and Figure 1(b)), which provides us with more comprehensive spatial information. Increasing the flight line overlap results in multiple angular sampling, higher point density and mitigates occlusions. Although the data density is still relatively low, compared with TLS, ULS can provide previously unseen overstory details due to the top-down view and overlap flight line. Furthermore, ULS is considered to be more suitable for conducting long-term monitoring of forests than TLS, as it allows predefined flight plans with minimal operator involvement.

Consequently, leaf-wood semantic segmentation in ULS data is required to accurately monitor foliage density variation over space and time. Changes in forest foliage density are indicative of forest functioning and their tracking may have multiple applications for carbon sequestration prediction, forest disease monitoring and harvest planning. Fulfilling these requirements necessitates the development of a robust algorithm that is capable to classify leaf and wood in forest environments. While numerous methods have demonstrated effective results on TLS data (see Section 2), these methods cannot be applied directly to ULS, due in particular to the class imbalance issue: leaf points account for about 95% of the data. Another problem is that many methods rely on the extra information provided by LiDAR devices, such as intensity. In the context of forest monitoring, intensity is not reliable due to frequent pulse fragmentation and variability in natural surface reflectivity (see Vincent et al. [5]). Furthermore, the reflectivity of the vegetation is itself affected by diurnal or seasonal changes in physical conditions, such as water content or leaf orientation (Brede et al. [3]). Therefore, methods relying on intensity information (Wu et al. [6]) may exhibit substantial

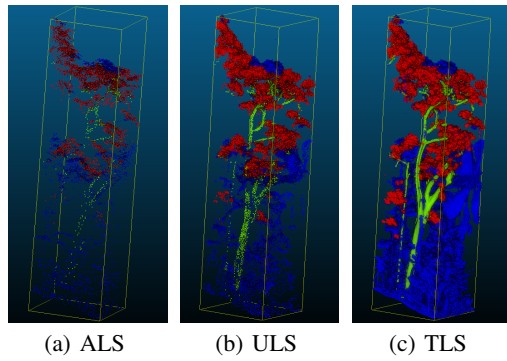

(a) ALS          (b) ULS          (c) TLS

Figure 1: Point clouds produced by three scanning modes on the same area (20m × 20m × 42m), illustrate how much the visibility of the understory differs. The colors in the figure correspond to different labels assigned to the points, where red and green indicate leaves and wood, respectively. Blue points are unprocessed, so labeled as unknown. TLS data were initially subjected to semantic segmentation using LeWos [4] and subsequently manually corrected. Next, the K-nearest neighbors (KNN) algorithm was used to assign labels to ALS and ULS data based on the majority label among their five nearest neighbors in TLS data.

variations in performance across different locations and even within the same location for different acquisition batches. To address this issue, certain methods (LeWos proposed by Wang et al. [4]; Morel et al. [7]) have good results while exclusively utilizing the spatial coordinates of LiDAR data.

Inspired by the existing methods, we propose a novel end-to-end approach **SOUL** (Semantic segmentation On ULs) based on PointNet++ proposed by Qi et al. [8] to perform semantic segmentation on ULS data. SOUL uses only point coordinates as input, aiming to be applicable to point clouds collected in forests from various locations worldwide and with sensors operating at different wavelengths. The foremost concern to be tackled is the acquisition of labeled ULS data. Since no such data set existed up to now, we gathered a ULS data set comprising 282 trees labeled as shown in Figure 4. This was achieved through semi-automatic segmentation of a coincident TLS point cloud and wood/leaf label transfer to ULS point cloud. Secondly, the complex nature of tropical forests necessitates the adoption of a data pre-partitioning scheme. While certain methods (Krisanski et al. [9]; Wu et al. [10]) employ coarse voxels with overlap, such an approach leads to a fragmented representation and incomplete preservation of the underlying geometric information. The heterogeneous distribution of points within each voxel, including points from different trees and clusters at voxel boundaries, poses difficulties for data standardization. We introduce a novel data preprocessing methodology named geodesic voxelization decomposition (GVD), which leverages geodesic distance as a metric for partitioning the forest data into components and uses the topological features, like intrinsic-extrinsic

ratio (IER) (He et al. [11]; Liu et al. [12]), to preserve the underlying geometric features at component level (see Section 1). The last issue concerns the class imbalance problem during the training stage. To address this issue, we developed a novel loss function named the rebalanced loss, which yielded improved performance compared with the focal loss (Lin et al. [13]) for our specific task. This enhancement resulted in a 23% increase in the ability to recognize wood points, see Table 1.

The contribution of our work is three-fold. First, SOUL is the first approach developed to tackle the challenge of semantic segmentation on tropical forest ULS point clouds. SOUL demonstrates better wood point classification in complex tropical forest environments while exclusively utilizing point coordinates as input. Experiments show that SOUL exhibits promising generalization capabilities, achieving good performance even on data sets from other LiDAR devices, with a particular emphasis on overstory. Secondly, we propose a novel data preprocessing method, GVD, used to pre-partition data and address the difficult challenge of training neural networks from sparse point clouds in tropical forest environments. Third, we mitigate the issue of imbalanced classes by proposing a new loss function, referred to as rebalanced loss function, which is easy to use and can work as a plug-and-play for various deep learning architectures. The data set (Bai et al. [14]) used in the article is already available in open access at https://zenodo.org/record/8398853 and our code is available at https://github.com/Na1an/phd_mission.

## 2   Related Work

**Classical methods.**   Classical methods are prevalent for processing point cloud data in forest environments, especially for semantic segmentation of TLS data, where they have achieved high levels of efficacy and applicability in practice. One such method, LeWos (Wang et al. [4]) uses geometric features and clustering algorithms, while an alternative method proposed by the first author of LeWos (Wang [15]) employs superpoint graphs (Landrieu & Simonovsky [16]). Moorthy et al. [17] use random forests (RF), and Zheng et al. [18] explore Gaussian mixture models. Other approaches, such as those proposed by Lalonde et al. [19], Itakura et al. [20], Vicari et al. [21], and Wan et al. [22], have also demonstrated effective semantic segmentation of TLS data.

**Deep Learning methods.**   Pioneering works like FSCT proposed by Krisanski et al. [9] and the method proposed by Morel et al. [7] have shown promising results by using PointNet++ [8]. Shen et al. [23] use PointCNN (Li et al. [24]) as the backbone model to solve the problem of segmenting tree point clouds in planted trees. Wu et al. [6] propose FWCNN, which incorporates intensity information and provides three new geometric features. Windrim & Bryson [10] propose a method that combines Faster R-CNN [25] and Pointnet [26], which utilizes the bird's eye view technique for single tree extraction and performs semantic segmentation on individual trees. In general, deep neural network-based approaches have demonstrated better performance in semantic segmentation of TLS point clouds compared to classical methods.

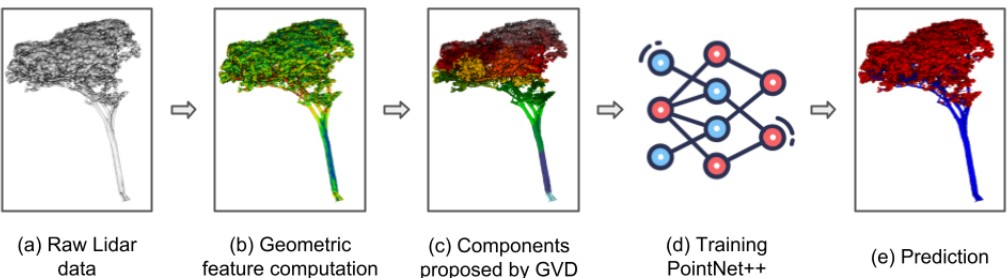

Figure 2: Overview of SOUL. (a) We use only the coordinates of raw LiDAR data as input. (b) Four geometric features linearity, sphericity, verticality, and PCA1 are calculated at three scales using eigenvalues, then standardized. (c) GVD proposes partitioned components and performs data normalization within these components. (d) Training deep neural network. (e) Finally, output are point-wise predictions.

# 3 SOUL: Semantic segmentation on ULS

SOUL is based on PointNet++ Multi-Scale Grouping (MSG) [8] with some adaptations. The selection of PointNet++ is not only because of its demonstrated performance in similar tasks (Krisanski et al. [9]; Morel et al. [7]; Windrim & Bryson[10]), but also because of the lower GPU requirements (Choe et al. [27]) compared with transformer-based models developed in recent years, like the method proposed by Zhao et al. [28]. The main idea of SOUL lies in leveraging a geometric approach to extract preliminary features from the raw point cloud, these features are then combined with normalized coordinates into a deep neural network to obtain more abstract features in some high dimensional space [26]. We will introduce our method in detail as follows.

## 3.1 Geometric feature computation

At this stage, we introduce four point-wise features: linearity, sphericity, verticality, and PCA1, which are computed at multiple scales of 0.3 m, 0.6 m, and 0.9 m in this task. To calculate these features, we need to construct the covariance matrix $\Sigma_{p_i}$ for each point $p_i$, compute its eigenvalues $\lambda_{1,i} > \lambda_{2,i} > \lambda_{3,i} > 0$ and the corresponding normalized eigenvectors $e_{1,i}$, $e_{2,i}$, $e_{3,i}$. The local neighbors $\mathcal{N}_{p_i}$ of $p_i$ are given by:

$$\mathcal{N}_{p_i} = \{q \mid q \in \mathcal{P}, d_e(p_i, q) \leq r\} \tag{1}$$

where $\mathcal{P}$ is the set of all points, $d_e$ is the Euclidean distance and $r \in \{0.3, 0.6, 0.9\}$.

The covariance matrix $\Sigma_{p_i}$ is:

$$\Sigma_{p_i} = \frac{1}{|\mathcal{N}_{p_i}|} \sum_{p_j \in \mathcal{N}_{p_i}} (p_j - \bar{p})(p_j - \bar{p})^T \tag{2}$$

where $\bar{p}$ is the barycenter of $\mathcal{N}_{p_i}$.

The equations for computing the four characteristics are as follows:

**Linearity**    $L_{p_i}$ serves as an indicator for the presence of a linear 1D structure (Weinmann et al. [29]).

$$L_{p_i} = \frac{\lambda_{1,i} - \lambda_{2,i}}{\lambda_{3,i}} \tag{3}$$

**Sphericity**    $S_{p_i}$ provides information regarding the presence of a volumetric 3D structure (Weinmann et al. [29]).

$$S_{p_i} = \frac{\lambda_{3,i}}{\lambda_{1,i}} \tag{4}$$

**Verticality**    $V_{p_i}$ indicates the property of being perpendicular to the horizon (Hackel et al. [30]), LeWos [4] uses it as a crucial feature because of its high sensitivity to the tree trunk. Here [0 0 1] is the vector of the canonical basis.

$$V_{p_i} = 1 - |\langle [0\ 0\ 1],\ e_{3,i} \rangle| \tag{5}$$

**PCA1**    $PCA1_{p_i}$ reflects the slenderness of $\mathcal{N}_{p_i}$. The direction of the first eigenvector $e_1$ is basically the most elongated in $\mathcal{N}_{p_i}$.

$$PCA1_{p_i} = \frac{\lambda_{1,i}}{\lambda_{1,i} + \lambda_{2,i} + \lambda_{3,i}} \tag{6}$$

Each point is endowed now with 12 features encompassing 3 scales, enabling a precise depiction of its representation within the local structure. These features are subsequently standardized and integrated into the model's input as an extension to point coordinates for the deep learning model. Refer to Supplementary Section E for single-scale and multi-scale ablation studies.

---

**Algorithm 1** Geodesic Voxelization Decomposition (GVD)

---

**Input:** Voxel grid $\mathcal{V}$;   threshold $\tau$ on $d_{gv}$;   threshold $\gamma$ on IER
**Output:** Voxel partition in $I$ components $\mathbf{C} = \{C_i, i = 1 : I\}$

 1:  $i \leftarrow 1$                                                                    ▷ Component id
 2:  **function** GVD($\mathcal{V}$, $\tau$, $\gamma$)
 3:     **while** $\mathcal{V} \neq \{\}$ **do**
 4:         $C_i \leftarrow \emptyset$, $ier \leftarrow 0$, $gd \leftarrow 0$
 5:         Choose $\hat{v} \in \mathcal{V}$
 6:         **while** $ier < \gamma$ and $gd < \tau$ **do**
 7:             Choose $\bar{v} \in \mathcal{C}(\hat{v})$                    ▷ $\mathcal{C}(\hat{v})$ arises from BFS on $\hat{v}$
 8:             $gd \leftarrow d_{gv}(\hat{v}, \bar{v})$
 9:             $ier \leftarrow IER(\hat{v}, \bar{v})$                                   ▷ Eq. (8)
10:             $C_i \leftarrow C_i \cup \{\bar{v}\}$
11:         **end while**
12:         $\mathbf{C} = \mathbf{C} \cup C_i$
13:         $i \leftarrow i + 1$
14:     **end while**
15:     **Return C**
16: **end function**

---

## 3.2   Data pre-partitioning

The GVD algorithm 1 is a spatial split scheme used to partition the ULS data while preserving the topology of the point cloud. This approach enables the extraction of a set of representative training samples from raw forest data, while preserving the local geometry information in its entirety. The point cloud data is first voxelized into a voxel grid $\mathcal{V}$ with a voxel size $s$ equal to 0.6 m, an empirical value, and equipped with a neighborhood system $\mathcal{C}$. Each voxel $v$ is considered as occupied if it contains at least one point. The geodesic-voxelization distance $d_{gv}$ between two voxels is defined as 1 if they are adjacent, and as the Manhattan distance between $v$ and $v'$ in voxel space otherwise. This can be expressed mathematically as:

$$d_{gv}(v, v') = \begin{cases} 1, & \text{if } v \text{ and } v' \text{ are adjacent in } \mathcal{C} \\ |x - x'| + |y - y'| + |z - z'|, & \text{otherwise} \end{cases} \qquad (7)$$

where $(x, y, z)$ and $(x', y', z')$ denote the integers coordinates referring to the index of $v$ and $v'$ respectively. We can establish an inter-voxel connectivity information with $d_{gv}$ that enables the computation of the intrinsic-extrinsic ratio (IER):

$$IER(v, v') = \frac{d_{gv}(v, v')}{d_e(v, v')} \ . \qquad (8)$$

We believe that $d_{gv}$ and $IER$ may provide strong cues for the subsequent segmentation task.

Subsequently, we choose one lowest voxel $\hat{v} \in \mathcal{V}$ on the vertical z-axis then employ the Breadth-first search (BFS) algorithm to explore its neighbors and the neighbors of neighbors $\mathcal{C}(\hat{v})$ and calculate the corresponding $d_{gv}(\hat{v}, \bar{v})$ and $IER(\hat{v}, \bar{v})$ where $\bar{v} \in \mathcal{C}(\hat{v})$. This procedure terminates when the geodesic-voxelization distance $d_{gv}$ exceeds $\tau$ or the $IER$ exceeds $\gamma$. $\tau$ is set to 10 and $\gamma$ is set to 1.5 in this task. All the points within $\mathcal{C}(\hat{v})$ are extracted and consolidated as a component $\mathbf{C}_i$ of the point clouds (see Figure 3(a) and Algorithm 1). The whole procedure is repeated until all the data is processed.

As the data is partitioned into multiple components, the points within one component $\mathbf{C}_i$ are further sorted based on their geodesic-voxelization distance. In this arrangement, points inside one voxel $v$ have the same geodesic distance $d_{gv}(\hat{v}, v)$, where $\hat{v}$ is the lowest voxel inside $\mathbf{C}_i$ (see Figure 3(b)). Thus, the GVD algorithm concludes at this stage. Moreover, minimum voxel number and minimum point number within a component are configurable, more details are provided in Supplementary Section C.

### 3.3 Normalization inside component

Shifting and scaling the point cloud coordinates before training a deep neural network is a widely adopted practice that can lead to improved training stability, generalization, and optimization efficiency, see Qi et al. [26]. Here, prior to training the model, the point coordinates are shifted to (0,0,0) and normalized by the longest axis among the three, all the point coordinates being confined within the (0,1) range. This prevents certain coordinates from dominating the learning process simply because they have larger values.

### 3.4 Deep neural network training

We employ a modified version of PointNet++ MSG for SOUL [8]. The data combination of points and features within a component is further partitioned into batches of 3,000, which serve as the input for the network. If a batch within a component has fewer than 3,000 points, the remaining points are randomly chosen from the same component. The labels of the leaves and wood points are 0 and 1, respectively. Define $B_k$ as such a batch, $|B_k| = 3,000$ and $B_k = B_{k,0} + B_{k,1}$ where $B_{k,0}$ and $B_{k,1}$ respectively represent the disjoint sub-batches in $B_k$ with ground-truth leaf points and wood points.

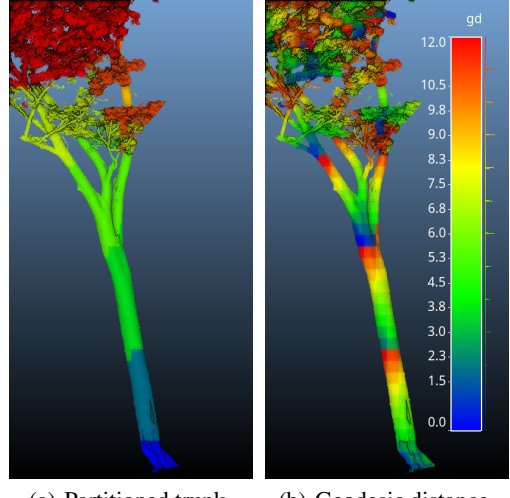

(a) Partitioned trunk      (b) Geodesic distance

Figure 3: (a) displays the components found by the GVD algorithm. (b) displays the geodesic distance within the corresponding component. The figures are illustrative, actual ULS data lacks such depicted complete tree trunks.

**Addressing class imbalance issue.** Within the labeled ULS training data set, only $4.4\%$ are wood points. The model is overwhelmed by the predominant features of leaves. Therefore, we propose a rebalanced loss $L_R$ that changes the ratio of data participating to 1:1 at the loss calculation stage by randomly selecting a number of leaf points equal to the number of wood points.

In practice, the rebalanced loss is:

$$L_R(Y_{B_k}) = -\sum y_k \log(\hat{p}_k) + (1 - y_k) \log(1 - \hat{p}_k), \ y_k \in (B_{k,0}^{'} \cup B_{k,1}). \tag{9}$$

where $Y_{B_k}$ specifies the ground truth labels of batch $B_k$, $\hat{p} \in [0,1]$ is the model-estimated probability for the label $y = 1$ and $B_{k,0}^{'}$ is defined as

$$B_{k,0}^{'} = \begin{cases} \text{downsampling}(B_{k,0}, |B_{k,1}|), & \text{if } |B_{k,0}| \geq |B_{k,1}| \\ B_{k,0}, & \text{otherwise.} \end{cases} \tag{10}$$

where the downsampling procedure consists of randomly and uniformly selecting a subset of size $|B_{k,1}|$ within $B_{k,0}$ without replacement.

The ablation study for the loss function can be found in Supplementary Section D.

## 4 Experiments

We first provide more details about various LiDAR data present in each data set. Secondly, the data preprocessing procedure is elaborated upon. Next, we delve into the specific configurations of model hyperparameters during the training process, along with the adaptations made to the PointNet++ architecture. Following that, we showcase the performance of SOUL on the labeled ULS data set. Finally, we demonstrate the generalization ability of SOUL on the other data sets.

### 4.1 Data Characteristics

The ULS data for this study were collected at the Research Station of Paracou in French Guiana (N5°18′ W52°53′). This tropical forest is situated in the lowlands of the Guiana Shield, the average height of the canopy is 27.2 m with top heights up to 44.8 m (Brede et al. [3]). Four flights of ULS data were collected at various times over the same site, with each flight covering 14,000 m² of land and consisting of approximately 10 million points, yielding a density of around 1,000 pts/m²[2]. In comparison, the point density of the labeled TLS data from the same site is about 320,000 pts/m². Further information pertaining to LiDAR scanners can be found in Supplementary Section B.

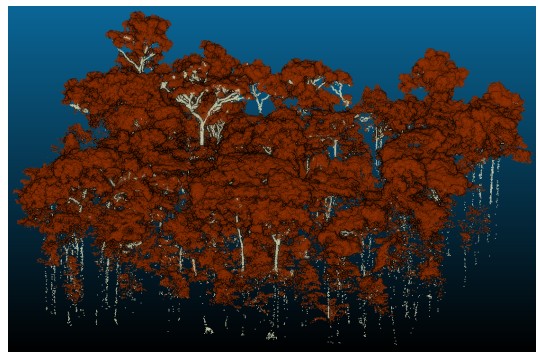

Figure 4: The labeled ULS data set in French Guiana, where only 4.4% of the points correspond to wood. This significant class imbalance of leaf & wood presents a considerable challenge in discriminating wood points from ULS forest point cloud.

In addition, we used two public data sets without labels only for exploring the model's generalization ability. One is a ULS data set from Sandhausen, Germany (Weiser et al. [31]), the other one is a mobile laser scanning (MLS) [3] data set from New South Wales (NSW), Australia (NSW & Gonzalez [32]). The Sandhausen data set uses the same equipment as we do, but exhibits a lower point density of about 160 pts/m². The point density of the NSW data set is approximately 53,000 pts/m².

## 4.2 Data preprocessing

The first step of the routine involves data cleaning for TLS and ULS. The points exhibiting an intensity lower than -20 dB and those displaying high deviations (Pfennigbauer & Ullrich [33]) are removed, subsequently the TLS data are subsampled by setting a 0.001 m coordinate precision. The next stage is to separate ground by applying a cloth simulation filter (CSF) proposed by Zhang et al. [34], which mimics the physical phenomenon of a virtual cloth draped over an inverted point cloud. The ULS data were subjected to a similar preprocessing pipeline, except that CSF on ULS (or ALS) performs poorly and was not used. Here we used the software TerraSolid ground detection algorithm on ALS and used the common ground model for all point clouds of the same plot.

## 4.3 Data annotation

A subset of TLS trees with large trunk is labeled using LeWos algorithm (Wang et al. [4]), followed by a manual refinement process (Martin-Ducup et al. [35]) to improve the outcome accuracy. Then the k-nearest neighbors algorithm (KNN) is employed to assign the tree identifiers (treeIDs) and the labels of TLS points to ULS points. Specifically, for each point in ULS data set, the KNN algorithm selects five nearest points in the TLS data set as reference points and subsequently determines the corresponding label based on the majority consensus of the reference points. It is noteworthy that within TLS data, there exist three distinct label types: unknown, leaf and wood, as demonstrated in Figure 1(c), which respectively correspond to labels: -1, 0, and 1. The unknown label is also transferred to ULS points, then such points are filtered out for the next step. Unlabeled and therefore excluded points represent 65% of the ULS point cloud.

To facilitate model training and quantitative analysis, the labeled ULS data was partitioned into training, validation, and test sets based on treeID, with 221 trees in the training data set, 20 in the validation data set, and 40 in the test data set. This partitioning is visually demonstrated in a video, where the trees are distinctly grouped together within each data set.

---

[2]Points per square meter.
[3]The LiDAR device is integrated into a backpack, enabling a similar scanning process akin to TLS.

## 4.4 Implementation details

The Adam optimization algorithm is chosen as the optimizer and the learning rate is 1e-7. Following the practice proposed by Smith et al. [36], we employ a strategy of gradually increasing the batch size by a factor of two approximately every 1,000 epochs until reaching a final batch size of 128 instead of decreasing the learning rate during the training process. According to our experience, achieving good results requires a substantial duration, typically exceeding 3,000 epochs. In this paper, the prediction results are obtained from a checkpoint saved at the 3161st epoch. At this epoch, SOUL achieved a Matthews Correlation Coefficient (MCC) value of 0.605 and an AUROC value of 0.888 on the validation data set.

The MCC expression is:

$$\mathbf{MCC} = \frac{TP \times TN - FP \times FN}{\sqrt{(TP + FP)(TP + FN)(TN + FP)(TN + FN)}} \qquad (11)$$

For binary classification, the calculation of the MCC metric uses all four quantities (TP, TN, FP and FN)[4] of the confusion matrix, and produces a high score only if the prediction is good for all quantities (see Yao & Shepperd [37]), proportionally both to the size of positive elements and the size of negative elements in the data set (Chicco et al. [38]).

In our task, an output in the form of a probability is preferred, as it is more consistent with the data collection process, since the LiDAR footprint yielding each single echo is likely to cover both wood and leaf elements. However, to facilitate performance comparison with other methods, the model generates a discrete outcome that assigns a label to each point. If the probability of a point being classified as a leaf exceeds the probability of being classified as wood, it is labeled as a leaf and vice versa. Additionally, the probabilistic outputs indicating the likelihood of being a leaf or wood point are retained. More details are provided in the Supplementary Section B.

## 4.5 Results on French Guiana ULS data

Comparatively to the prevailing methods employed for forest point clouds, our SOUL approach improves semantic segmentation on ULS forest data by large margins. The results are summarized in Table 1, while Figure 5 highlights the performance within the tree canopies. In terms of quantitative analysis, SOUL demonstrates a specificity metric of 63% in discerning sparse wooden points, where the metric specificity stands for the ratio of true wood points predicted as wood. This value can be elevated to 79%, but at the cost of decreasing the accuracy metric to 80%, where the metric accuracy stands for the ratio of correctly predicted wood and leaf points. For better understanding how each class

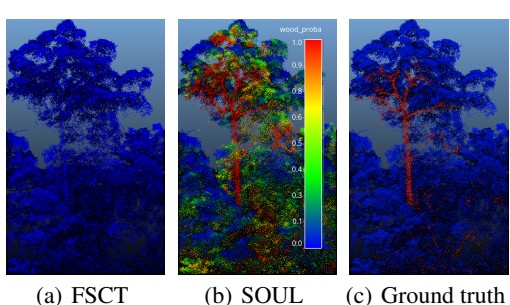

(a) FSCT    (b) SOUL    (c) Ground truth

Figure 5: Qualitative results on ULS test data. Because of the class imbalance issue, methods such as FSCT[9], LeWos[4], and other existing approaches developed for dense forest point clouds, like TLS or MLS, (cf. Section 2) are ineffective.

contributes to the overall performance, we introduced two more informative metrics mentioned in the study of Branco et al. [39], Geometric mean (G-mean) and Balanced Accuracy (BA). These metrics validate the performance of the SOUL model, providing more comprehensive insights into its capabilities.

## 4.6 Results on the other data.

After testing on data sets from Sandhausen data set (Weiser et al. [31]) and NSW Forest data set [32], we observed that the SOUL exhibits good generalization capability across various forests. Qualitative results are shown in Figure 6. First, SOUL demonstrates good results on Sandhausen data set [31]. However, SOUL does not outperform the others methods on TLS, but it shows better discrimination within the tree canopy. A possible improvement of SOUL performance is foreseen for denser data provided the model is retrained on similar data sets.

---

[4]TP, TN, FP, FN stand respectively True Positives, True Negatives, False Positives, False Negatives.

# 5 Discussion

**GVD as a spatial split scheme.**    Semantic segmentation in tropical forest environments presents distinctive challenges that differ from objects with regular shapes found in data sets such as ShapeNet (Chang et al. [40]) and ModelNet40 (Wu et al. [41]). Forest point clouds are inherently poorly structured and irregular because leaves and branches are small with regard to the LiDAR footprint size, a lot of points derived from LiDAR device are mixed between the two materials of leaves and wood. We propose GVD, an ad-hoc method for ULS data, to overcome the chaotic nature of forest point clouds with a pre-partition step. Similar to the enhancement that RCNN (Girshick et al. [42]) brings to CNN (LeCun et al. [43]) on image classification and semantic segmentation tasks, GVD serves as a region proposal method that provides improved training sample partitioning for deep learning networks. The neural network can benefit from more refined and informative data, leading to enhanced performance in the semantic segmentation task.

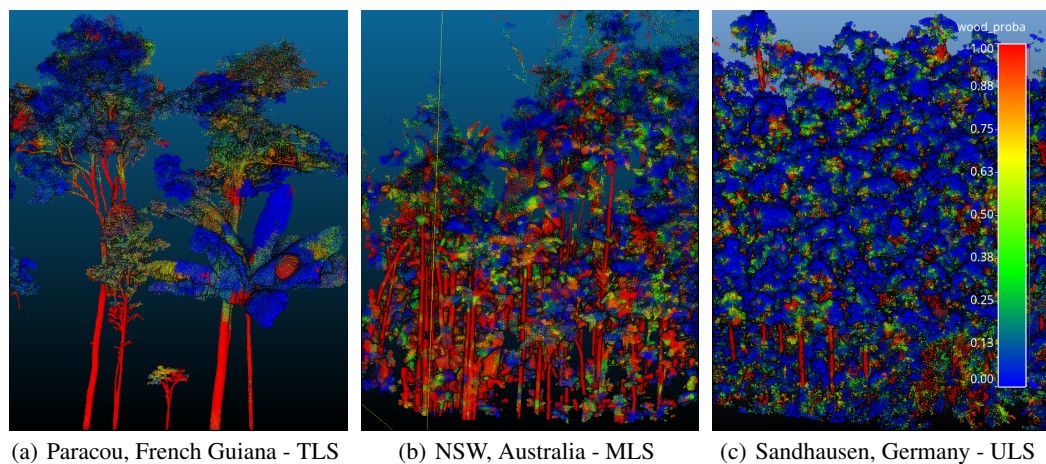

(a) Paracou, French Guiana - TLS   (b) NSW, Australia - MLS   (c) Sandhausen, Germany - ULS

Figure 6: Qualitative results on various LiDAR data from different sites.

**Rebalanced loss designed for class imbalance issue.**    In our model, PointNet++ is chosen as the backbone network due to its known robustness and effectiveness in extracting features from point clouds (Qian et al. [44]). But learning from imbalanced data in point cloud segmentation poses a persistent challenge, as discussed by Guo et al. [45]. To address this issue, we employ a rebalanced loss function tailored to this task. Rather than adjusting class weights during training (Lin et al. [13]; Sudre et al. [46]), we adopt a random selection approach to balance the training data during the loss computation. This decision stems from the observation that misprediction of a subset of the minority class can lead to significant fluctuations in the loss value, consequently inducing drastic changes in gradients. In an extreme scenario, wherein a single point represents a positive sample, correct prediction of this point drives the loss towards zero, irrespective of other point predictions. Conversely, misprediction of the single point yields a loss value approaching one.

Table 1: Comparison of different methods

| Methods | Accuracy | Recall | Precision | Specificity | G-mean | BA[1] |
|---|---|---|---|---|---|---|
| FSCT [9] | 0.974 | 0.977 | **0.997** | 0.13 | 0.356 | 0.554 |
| FSCT + retrain | **0.977** | **1.0** | 0.977 | 0.01 | 0.1 | 0.505 |
| LeWos [4] | 0.947 | 0.97 | 0.975 | 0.069 | 0.259 | 0.520 |
| LeWos (SoD[2]) [22] | 0.953 | 0.977 | 0.975 | 0.069 | 0.260 | 0.523 |
| SOUL (focal loss [13]) | 0.942 | 0.958 | 0.982 | 0.395 | 0.615 | 0.677 |
| SOUL (rebalanced loss) | 0.826 | 0.884 | 0.99 | **0.631** | **0.744** | **0.757** |

[1] BA (Balanced Accuracy) $BA = \frac{1}{2}(Recall + Specificity)$.

[2] SoD (Significance of Difference).

**SOUL model's generalization capability.** Despite being originally developed for tropical forest environments, SOUL demonstrates promising performance when applied to less complex forests. We anticipate that with the increasing prevalence of ULS data, coupled with more high-quality labeled data sets, the performance of SOUL is poised to advance further. Additionally, there is potential for SOUL to be extended to other complicated LiDAR scenes. It is conceivable to envision the development of a universal framework based on SOUL that can effectively handle various types of forest LiDAR data, including ULS, TLS, MLS, and even ALS.

**Ablation study of geometric features.** An ablation study is conducted to evaluate the impact of individual geometric features proposed in our model. Comparative experiments, as detailed in Figure 7 and Table 2, illustrate the advantages of utilizing multiple precomputed geometric features at various scales. Specifically, we observe that features like linearity and verticality significantly enhance trunk recognition, while PCA1 and Sphericity are effective within the canopy. While PointNet++ possesses the capability to internally learn complex features, it may not precisely replicate our empirically selected geometric attributes. PointNet++ excels at adapting and acquiring sophisticated features from training data, whereas our predefined geometric features offer specific advantages in our model's context.

Table 2: Comparison of single geometric feature and multiple geometric features

| Methods | Accuracy | Recall | Precision | Specificity | G-mean | BA[1] |
|---|---|---|---|---|---|---|
| SOUL (linearity + one scale) | **0.962** | **0.978** | 0.983 | 0.31 | 0.55 | 0.636 |
| SOUL (unabridged) | 0.826 | 0.884 | **0.99** | **0.631** | **0.744** | **0.757** |

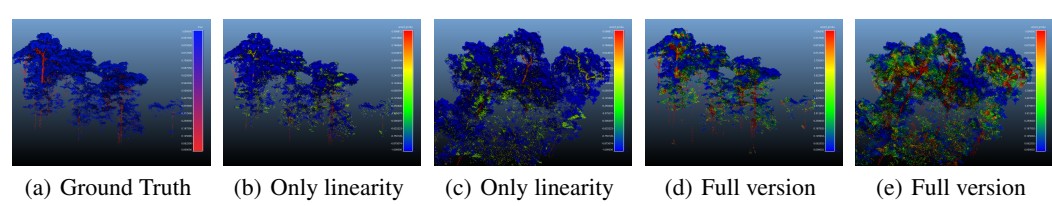

(a) Ground Truth    (b) Only linearity    (c) Only linearity    (d) Full version    (e) Full version

Figure 7: Comparing the unabridged model to a downgraded model using a single-scale feature linearity. Figures (c) and (e) represent different perspectives of Figures (a), (b), and (d).

**Limitations.** The current observation is that SOUL may not perform as well as other methods on TLS data, as it was not trained for that. Additionally, SOUL may not perform effectively for trees that significantly deviate from the training data. By incorporating diverse samples during training, the model can mitigate the challenges associated with vegetation disparity.

# 6 Conclusion

We present SOUL, a novel approach for semantic segmentation in complex forest environments. It outperforms existing methods in the semantic segmentation of ULS tropical forest point clouds and demonstrates high performance metrics on labeled ULS data and generalization capability across various forest data sets. The proposed GVD method is introduced as a spatial split schema to provide refined training samples through pre-partition. One key aspect of SOUL is the use of the rebalanced loss function, which prevents drastic changes in gradients and improves segmentation accuracy. While SOUL shows good performance for different forest types, it may struggle with significantly different trees without retraining. Future work can focus on improving the performance of SOUL on denser forest point clouds to broaden its applications.

## Acknowledgement and Disclosure of Funding

The authors are grateful to Nicolas Barbier and Olivier Martin-Ducup for their contributions in data collection and TLS data preprocessing. We also acknowledge that this work has been partially supported by MIAI@Grenoble Alpes, (ANR-19-P3IA-0003).

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

# Supplementary

## A  Overview

In this supplementary material, we provide more quantitative results, technical details and additional qualitative test examples. Section B presents more implementation and training details of SOUL, in Section C the efficacy of the GVD algorithm is discussed, while Section D conducts ablation studies to investigate the impact of different loss functions on the model's performance. At last, Section E demonstrates the performance improvement achieved by employing multiple-scale computation of geometric features compared to a single-scale approach. A video showcasing the performance of SOUL is available at this link: here.

## B  Additional Implementation Details

As a supplement to Sections 3.4 and 4.4, we provide additional details here.

**LiDAR scanner.**  Table 3 presents a summary of the distinguishing laser characteristics between ULS and TLS. The acquisition of LiDAR data is profoundly impacted by atmospheric characteristics, the LiDAR extinction coefficients exhibit a low impact to atmospheric humidity at both 905 nm and 1550 nm wavelengths (Wojtanowski et al. [47]). ULS uses 905 nm as wavelength, at which the scanning device hast the minimum energy consumption (Brede et al. [3]). Meanwhile, the TLS system operates at a wavelength of 1550 nm, which is more susceptible to the fog impact (Wojtanowski et al. [47]). This poses the issue that the spectral reflectance of leaf and wood is less contrasted (Brede et al. [3]) at the 905 nm wavelength, thereby leading us to employ only point coordinates as input for avoiding the use of intensity information.

Table 3: Laser sensor characteristics

|                              | TLS        | ULS          |
|------------------------------|------------|--------------|
| RIEGL Scanner                | VZ400      | miniVux-1UAV |
| Laser Wavelength (nm)        | 1550       | 905          |
| Beam divergence (mrad)       | $\leq 0.25$ | $\leq 1.6*0.5$ |
| Footprint diameter (cm@100m) | 3.5        | $16*5$       |
| Pulse duration (ns)          | 3          | 6            |
| Range resolution (m)         | 0.45       | 0.9          |

**Training details.**  In addition to the content already illustrated in the main paper, we provide further details regarding the training parameters. First and foremost, sufficient training time is required for the model to achieve desirable performance. Our observation indicates that models trained less than 2,000 epochs are inadequately trained to achieve desirable performance. During the training process, three metrics, AUROC, MCC and Specificity, are considered more informative and insightful. Especially, the metric of specificity is crucial as it measures the ability to accurately discriminate wooden points, which is the primary requirement of our method. Another thing to note is that, contrary to the mentioned practice in the main text of increasing the batch size by a factor of two every 1,000 epochs, we did not strictly follow this restriction during the actual training process. Often, the increase in batch size occurred around 800-900 or 1100-1200 epochs for better using the GPU resources. However, theoretically, this offset should not affect the final performance.

**Architecture of DL model.**  We have made several modifications to the architecture of PointNet++ (MSG) [8]. Firstly, we observed that Adam (Kingma & Ba [48]) outperformed Nesterov SGD (Liu & Belkin [49]) in this task, and ELU (Clevert et al. [50]) activation function was better than ReLU (Nair & Hinton [51]). The fraction of the input units to drop for dropout layer was changed from 0.5 to 0.3, that means the probability of an element to be zeroed is 30% (Paszke et al. [52]). We also decreased the number of hidden neurons and added two more fully connected layers at the end.

**Test with error bar.**  We have calculated a confidence interval at 95% confidence level to indicate the uncertainty of our method. In the main body of the paper, it was mentioned that we obtained

data from four flights, and each data collection from these flights allowed us to obtain a labeled ULS data set through label transfer from TLS data. The test data set is composed by 40 trees from the same positions across these four labeled ULS data sets, resulting in a total of 160 trees. We divided further these 160 trees based on their unique treeID to 16 smaller test sets. However, trees with the same treeID are not placed together in the same test set. The calculation of confidence intervals was derived from these 16 test sets. The result is summarized in Table 4 and Table 5. It is clear that SOUL outperforms all other methods on ULS data and achieves state-of-the-art performance on metrics such as Specificity, Balanced Accuracy and G-mean.

Table 4: Comparison of different methods

| Methods | Specificity | G-mean | BA[1] |
|---|---|---|---|
| FSCT [9] | 0.13 | 0.356 | 0.554 |
| FSCT + retrain | 0.01 | 0.1 | 0.505 |
| LeWos [4] | 0.069 | 0.259 | 0.520 |
| LeWos (SoD[2]) [22] | 0.069 | 0.260 | 0.523 |
| SOUL (focal loss [13]) | 0.395 | 0.615 | 0.677 |
| SOUL (rebalanced loss) | **0.576 ± 0.063** | **0.651 ± 0.030** | **0.720 ± 0.027** |

[1] BA (Balanced Accuracy) $BA = \frac{1}{2}(Recall + Specificity)$.
[2] SoD (Significance of Difference).

Table 5: Comparison of different methods

| Methods | Accuracy | Recall | Precision |
|---|---|---|---|
| FSCT [9] | 0.974 | 0.977 | 0.997 |
| FSCT + retrain | 0.977 | 1.0 | 0.977 |
| LeWos [4] | 0.947 | 0.97 | 0.975 |
| LeWos (SoD) [22] | 0.953 | 0.977 | 0.975 |
| SOUL (focal loss [13]) | 0.942 | 0.958 | 0.982 |
| SOUL (rebalanced loss) | 0.857 ± 0.014 | 0.865 ± 0.015 | 0.988 ± 0.002 |

**Output contains redundant points.** To proceed with further data analysis and usage, it is necessary to remove duplicate points present in the output.

## C  Assessing the efficacy of GVD

The ULS data often covers several hectares, or even hundreds of hectares. The situation in tropical forests is also highly complex, with various types and sizes of trees densely packed together. The significant memory demand makes it nearly impossible to process all the data at once, leading us to adopt a spatial split scheme approach as a necessary compromise.

We can select data randomly from the whole scene, but selecting data randomly can result in a sparse and information-poor sample. An alternative is to employ a divide and conquer strategy to handle the chaotic, big volume, and complex ULS data. That is why we propose GVD, a method that involves breaking down the data into more manageable subsets (see Figure 8), allowing us to handle the intricacies and extract meaningful insights in a more systematic manner. This approach enables us to retain the information-rich aspects of the data while overcoming computational challenges associated with the sheer volume of data.

Prior to employing the GVD method, we initially adopted a more intuitive approach. The data was partitioned in unit of voxel, serving as component for batch preparation through down-sampling. However, this approach gave rise to border effects, particularly impeding SOUL's focus on the meticulous segmentation of intricate branch and leaf within tree canopy (see Figure 9(c)). The segmentation of cubes led to the emergence of noise point clusters along the voxel edges. ULS have more points on tree canopy, so the presence of noise point clusters on voxel edges is more severe on tree canopy, which imposes bigger obstacles to our leaf/wood segmentation task.

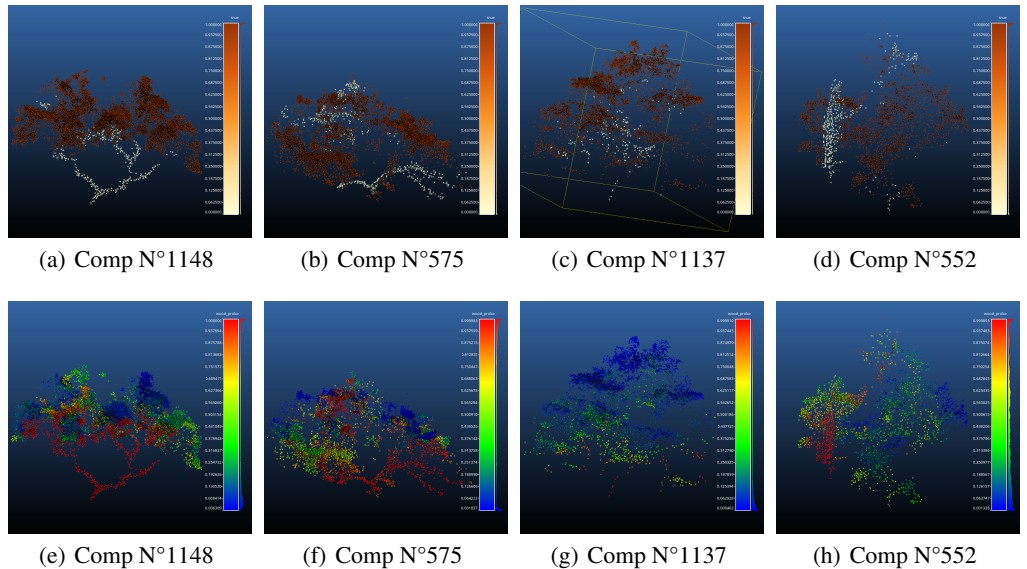

| (a) Comp N°1148 | (b) Comp N°575 | (c) Comp N°1137 | (d) Comp N°552 |
| (e) Comp N°1148 | (f) Comp N°575 | (g) Comp N°1137 | (h) Comp N°552 |

Figure 8: The figure depict distinct samples subjected to GVD processing and presents qualitative results within each sample. The upper row illustrates the ground truth for each sample (where brown represents leaves and white represents wood), whereas the lower row exhibits the predictive results generated by SOUL (where blue represents leaves and red represents wood incorporating gradient colors for transitional probability).

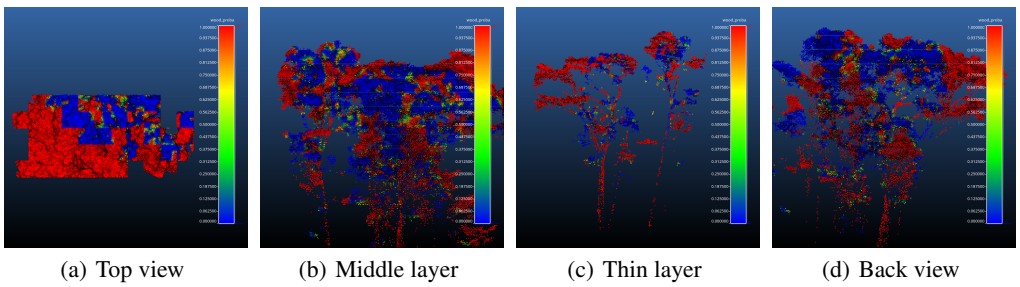

| (a) Top view | (b) Middle layer | (c) Thin layer | (d) Back view |

Figure 9: Downgrade version's output with sliding window. The use of GVD eliminates observed edge effects in this context.

We experimented with cuboids, a choice aimed at preserving greater semantic information within each component sample and expanding the spatial range for batch selection. Similar to a "sliding window", we can systematically traverse the entire forest with overlapping coverage in this way. But border effects persisted (see Figure 9), prompting the introduction of the GVD method, which led to a substantial improvement. Modifying parameters $\tau$ and $\gamma$ adjusts the coverage scope of each component in GVD segmentation. Additionally, tweaking the minimum accepted number of voxels and points, two GVD's configurable thresholds, within each component effectively manages component size. This process aids in the elimination of low-information content outlier point clusters to a certain extent.

A comparison between the result of the downgraded version using sliding window as spatial split schema and the individual-component point performance of full version SOUL is illustrated in Figure 8. Notably, in component N°552, SOUL effectively discriminates trunk points from leaf points that were previously entirely intertwined, surpassing our initial expectations.

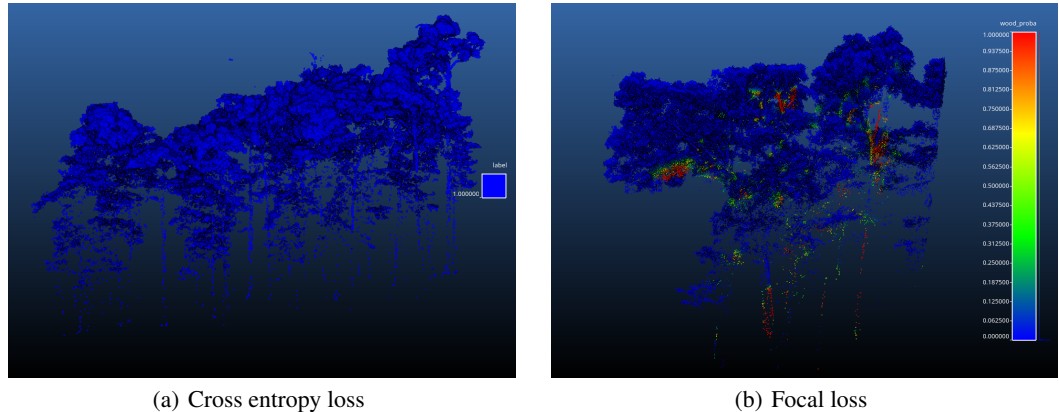

(a) Cross entropy loss            (b) Focal loss

Figure 10: The predictions of SOUL model on raw ULS data training with cross entropy loss function and focal loss function. Blue represents wood points with high probability, while red represents leaf points with high probability. Despite utilizing the focal loss, the model remains overwhelmed by leaf points.

## D    Ablation study of rebalanced loss

In this section, we mainly discuss the significant benefits introduced by the rebalanced loss. The network is biased towards leaf points when using cross-entropy loss function and the use of focal loss function (Lin et al. [13]) does not yield substantial improvements to the task, as its performance remains unsatisfactory when tested on the raw data set (see Figure 10(a) and Figure 10(b)).

Through a comparison of the specificity curves (see focal loss specificity in Figure 11 versus rebalanced loss specificity in Figure 12) and the training/validation loss curves of focal loss and rebalanced loss (see loss curve of the focal loss in Figure 13 versus loss curve of the rebalanced loss in Figure 14), we observed that focal loss failed to address the issue of class imbalance in our task. Plus, it is important to note that in the loss curve of rebalanced loss, the validation loss curve showed significant fluctuations in Figure 14. As foreseen, this variation occurred because the training process employed the rebalanced loss for model training, while the validation process utilized the cross-entropy loss function. Consequently, the loss value used for backpropagation is derived from the rebalanced loss function, which inherently does not consider the majority of leaf points. So once using cross entropy to calculate the loss, all points within a batch are taken into account, which can lead to fluctuations in the validation loss curve. However, it was expected that the validation loss curve would eventually converge on the validation set to demonstrate that using rebalanced loss for backward propagation effectively balances the representation of leaf and wood points. In fact, the final results surpassed focal loss by a significant margin, as evidenced by the convergence of the validation loss curve for rebalanced loss (see Figure 14). For the loss curve of focal loss in Figure 13, both the training and validation processes employ the focal loss function.

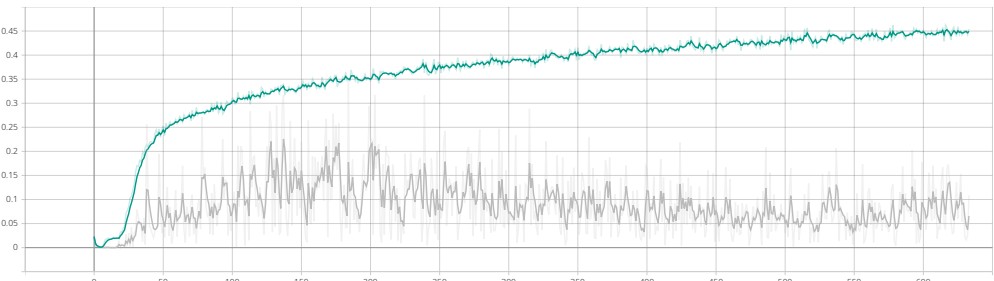

Figure 11: Specificity - focal loss (single-scale feature calculation). In the figure, the cyan curve represents the specificity curve of the training data set, while the gray curve represents the specificity curve of the validation data set.

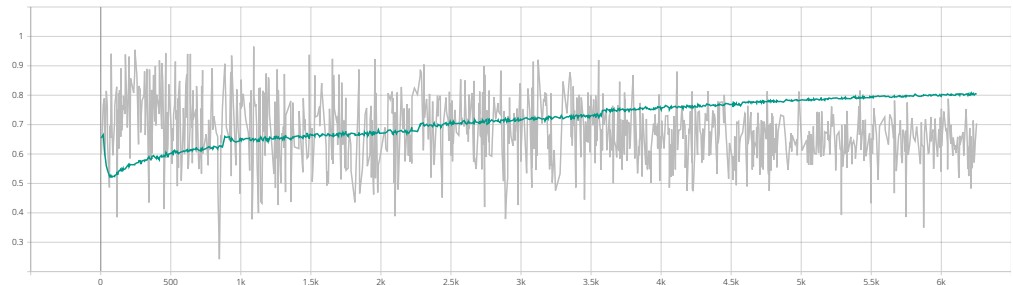

Figure 12: Specificity - rebalanced loss (single-scale feature calculation), cyan curve - training data set, gray curve - validation data set.

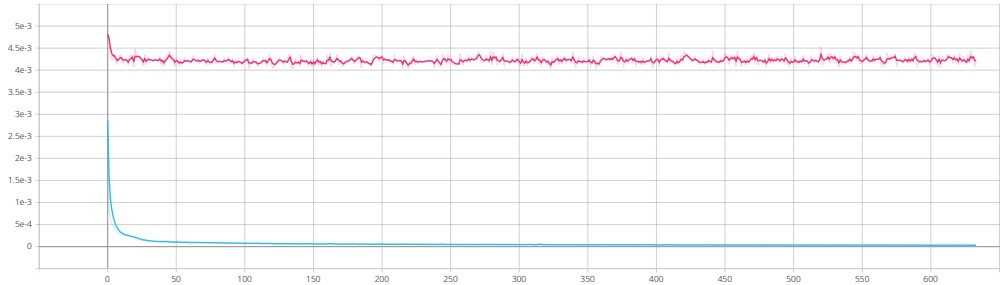

Figure 13: Loss - focal loss (single-scale feature calculation). The blue curve represents the loss curve of the training data set, while the red curve represents the loss curve of the validation data set.

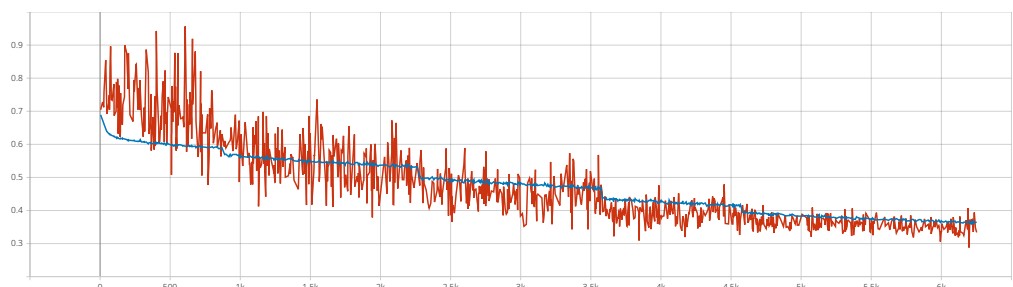

Figure 14: Loss - rebalanced loss (single-scale feature calculation), blue curve - training data set, red curve - validation data set.

In Figure 12, Figure 14, Figure 15, Figure 16 and Figure 17, we can clearly observe the presence of "sharp fluctuations", which are a result of the operation mentioned on main paper and Section B, where we increased the batch size by a factor of two approximately every 1,000 epochs. We followed this practice proposed by Smith et al. [36], which involves increasing the batch size instead of decaying the learning rate.

## E    Single-scale vs Multiple-scale

In this section, we mainly showcase multiple-scale geometric features calculation outperform single-scale at our task. After applying the rebalanced loss as the loss function, we observed that computing geometric features at multiple scales ultimately improved the performance of SOUL. When comparing the evolution of specificity with the number of epochs between single-scale and multiple-scale (see Figure 16 and Figure 17), multiple-scales not only exhibit higher values but also converge more effectively over time, same for loss convergence (see Figure 14 and Figure 15).

The Matthews Correlation Coefficient (MCC) (see Yao & Shepperd [37]) and AUROC are both effective metrics to evaluate the model performance under class imbalance. Therefore, besides the specificity, we provide the MCC and AUROC values for both single-scale and multiple-scale cases

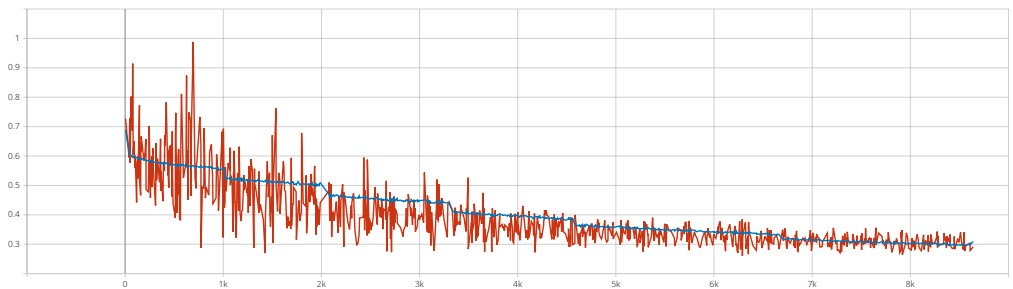

Figure 15: Loss - rebalanced loss (multiple-scale feature calculation), blue curve - training data set, red curve - validation data set.

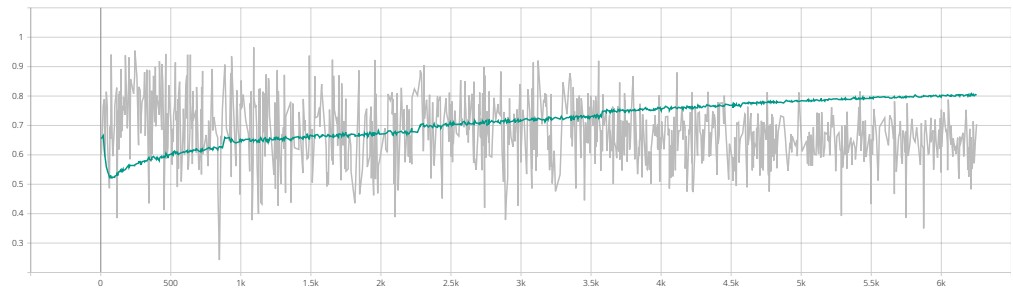

Figure 16: Specificity - Single-scale geometric features calculation, cyan curve - training data set, gray curve - validation data set.

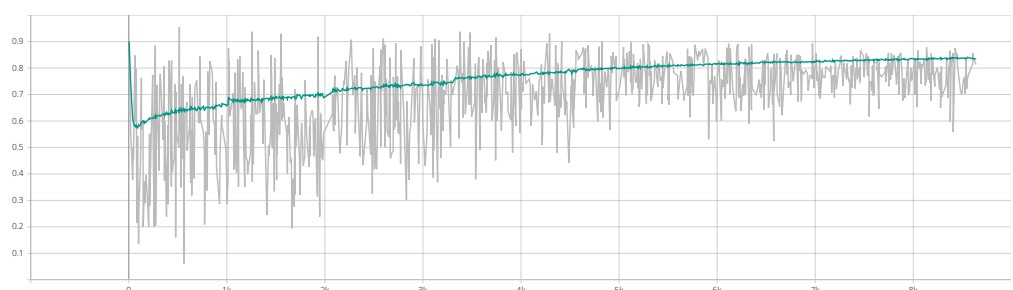

Figure 17: Specificity - Multiple-scale geometric features calculation, cyan curve - training data set, gray curve - validation data set.

during the training process. Upon analyzing the comparative results of MCC in Figure 18 and Figure 19, and AUROC in Figure 20 and Figure 21, we consistently observe that the multiple-scale computation of geometric features output higher values compared to the single-scale. This strongly supports the superior performance of multiple-scale feature computation over single-scale.

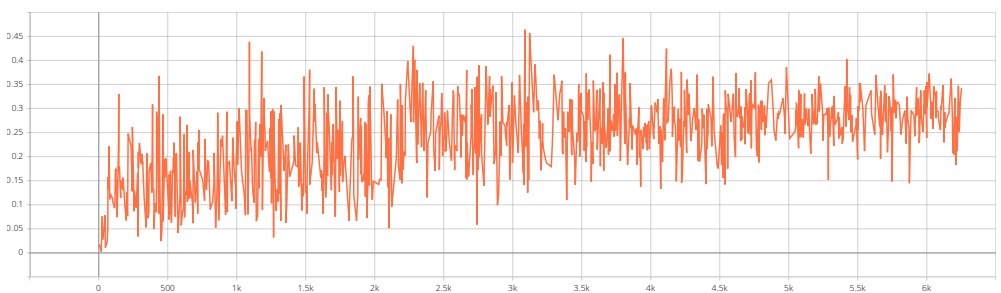

Figure 18: During the training process, the MCC (single-scale) values change with the number of epochs.

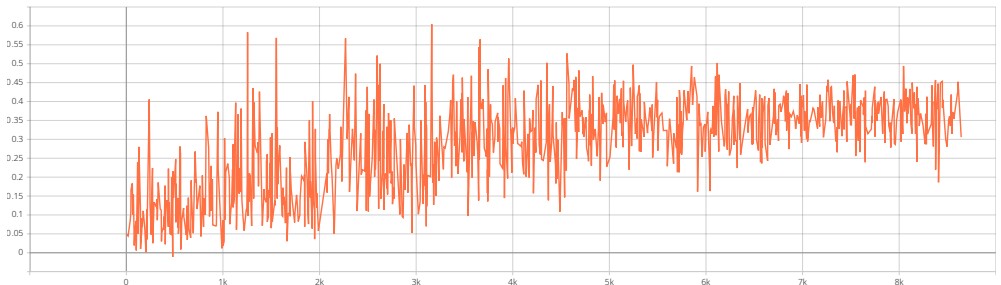

Figure 19: MCC (multiple-scale) values change with the number of epochs.

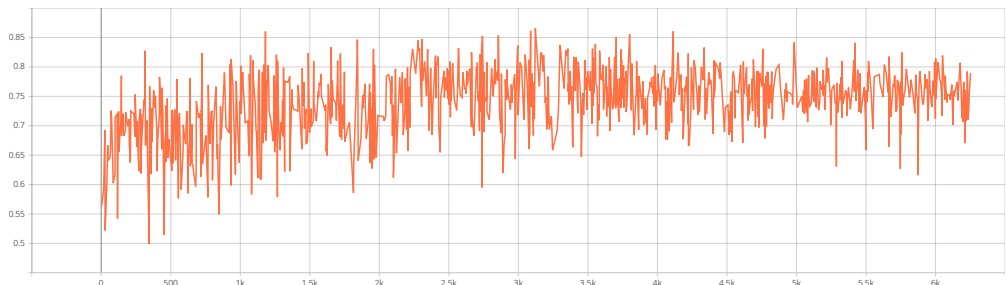

Figure 20: AUROC (single-scale) values change with the number of epochs.

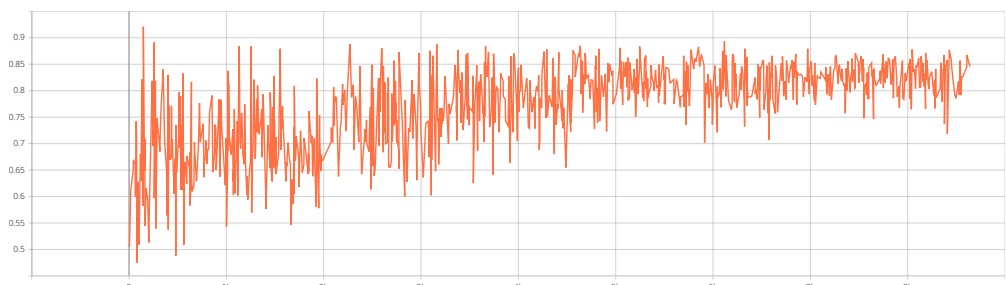

Figure 21: AUROC (single-scale) values change with the number of epochs.

MCC and AUROC provide valuable insights for selecting the final model state (checkpoint). For example, up to now, the best-performing model state is the one obtained at the 3161st epoch with the multiple-scale geometric features computation, as mentioned in the main paper. The model achieves an MCC value of 0.605 and an AUROC value of 0.888, these results generally outperform all single-scale metrics. As MCC is a discrete case of Pearson correlation coefficient, we can conclude that there exists a strong positive relationship exists between the model's predictions and the ground truth by using the interpretation of Pearson correlation coefficient.

