# OpenReview forum: "Semantic segmentation of sparse irregular point clouds for leaf/wood discrimination"
_NeurIPS.cc/2023/Conference — NeurIPS 2023 poster_

### Official Review · Reviewer_bCLj · 2023-06-20

**Soundness:** 2 fair
**Presentation:** 2 fair
**Contribution:** 2 fair
**Rating:** 6
**Confidence:** 5

**Summary:**

The authors proposed a method for semantic segmentation applied to unmanned aerial vehicle (UAV) laser scanning (ULS), namely SOUL, to discriminate leaf from wood points. It is based on PointNet++ with an additional sampling scheme and an innovative training loss function to handle the high imbalance between the classes. The SOUL method relies on the coordinates of the points to increase its range of application to other forests and other sensors. It also includes 4 point-wise geometric features computed at 3 scales to characterize each point.
The geodesic voxelization decomposition (GVD) is also introduced as a preprocessing method to partition the ULS data while preserving the topology of the point cloud.
Experiments have been conducted on a dataset recorded in a French Guiana tropical forest. The proposed method has reached the best results on 3 over 6 evaluation metrics, including metrics adapted to unbalanced datasets.
The approach has also been qualitatively tested on open source datasets recorded in Australia and Germany with open source datasets showing a potential generalization on other forests and other LiDAR sensors.

**Strengths:**

1/ The application of semantic segmentation to LiDAR recordings of forests is a high priority for climate change and global warming understanding and mitigation. This original work could have a potentially high impact for reforestation/afforestation monitoring and carbon stock estimation.

2/ The proposed SOUL method is the first to be adapted to the density and unbalance of ULS forest recordings. The proposed GVD preprocessing method is also relevant for other LiDAR sensors and type of forests.

3/ The authors have successfully performed experiments on a datasets recorded in a tropical forest in French Guiana, showing best results compared to a few competing methods and according to metrics specifically adapted to unbalanced datasets. Additional qualitative experiments conducted on unannotated datasets have shown a potential generalization on other forests and other LiDAR sensors.

4/ The paper is well written and motivated with relevant arguments.

**Weaknesses:**

1/ There is a lack of related works and comparisons using other point cloud architectures which could have led to better performances [1, 2]. The selection of PointNet++ has been motivated by L127 "the lower GPU requirements compared with transformer-based models developed in recent years". The model should be selected as a trade off between the GPU consumption and the performances of different architecture. These experiments would have been appreciated to support the argument.

2/ An ablation study of the proposed geometric features would have been appreciated. What is the impact of each proposed feature?  What is the actual impact of these features against using the raw point cloud? Is a model such as PointNet++ capable of estimating these geometric characteristics internally?

3/ Even if the LiDAR point cloud is affected by atmospheric characteristics, it would have been interesting to see the performances of the proposed method using the reflectance as an additional feature per point.

4/ The lack of experiments, in particular with competing methods, ablation studies and standard deviations (see next section), makes the submission questionable as to the significance of the results.

[1] Y. Guo et al., Deep Learning for 3D Point Clouds: A Survey. In TPAMI 2020.

[2] B. Fei et al., Comprehensive Review of Deep Learning-Based 3D Point Cloud Completion Processing and Analysis. In TITS 2022.

**Questions:**

Questions:

1/ Will the dataset be release publicly? Even though the code is available, the only numerical results are presented on the ULS dataset in French Guiana. For reproducibility reasons (both training and testing), the final rate of this submission will be conditioned to the release of the dataset which is not mentioned in the paper. Note that it would be a noticeable contribution since it will be the only annotated ULS dataset for semantic segmentation.

2/ Footnote 3, P7 "MCC is a very good metric for evaluating binary classifier against class imbalance issue." What "very good" means  in this context?

3/ It is not perfectly clear how the annotations of the French Guiana forest have been built. Do you have any additional comment on this topic?


Comments:

1/ There is no reference made in the core text to the appendices, and vice versa, making the bridge difficult for the reader. These documents should not be considered independently, both should refer to each other.

2/ A focus of qualitative and quantitative results at the limits of the branches would have been interesting to conduct since it concentrates most of the uncertainties. It would show that the model is not learning just a smart threshold on the z axis to discriminate the trunk from the leaves.

3/ Please be more specific than "partitioned" (L227) to qualify a spatial split. This information is only clearly available in the video of the Appendices.

4/ Typo: it would be nice to have the full name of the algorithm appearing in the title of Algorithm 1 in P5.

5/ Table 1: if the top 1 results are in bold per columns, all top 1 results should be highlighted, included results from competing methods.

6/ Typo: L290 "Quantitative results demonstrated are shown in Figure 6." These are qualitative results.

7/ Appendices L43 "It is evident 43 that SOUL outperforms all other methods on ULS data and achieves state-of-the-art performance on 44 metrics such as Specificity, Balanced Accuracy and G-mean." It is not straightforward since the standard deviations for the competing methods have not been provided, neither in Table 2 nor in Table 3 (Appendices). It is not possible to evaluate if the variance of the other methods include the average performance of the proposed one. Note that the methodology used to create the intervals are not usual in this submission, but still relevant. A more common practice is to estimate the standard deviation of the performances of several models with different initializations of the network.


**Limitations:**

The limits of the presented method has been addressed by the authors. The potential negative societal impacts have not been mentioned (e.g. military applications, UAV surveillance). The release of the dataset has not been mentioned which is a major limitation since the proposed method has been designed and validated on it and the only numerical results are based on its annotations.

---

> ### Author Rebuttal · Authors · 2023-08-04
>
> Thank you for your acknowledgment of our effort and the very helpful comments.
>
> Q1: Will the dataset be release publicly? ...
>
> We will release the labelled ULS data publicly along with the SOUL code, since this kind of data is
> still extremely rare indeed. It should make a useful contribution to the community of experts working
> on point cloud semantic segmentation of natural environments.
>
> Q2: For metric MCC, what "very good" means in this context?
>
> MCC = (TP * TN - FP * FN) / sqrt((TP + FP) * (TP + FN) * (TN + FP) * (TN + FN))
>
> For binary classification, the calculation of the Matthews correlation coefficient (MCC) metric uses
> all the four quantities (TP, TN, FP and FN) of the confusion matrix, and produces a high score only if
> the prediction obtained good results in all the four confusion matrix quantities[1][ 2], proportionally
> both to the size of positive elements and the size of negative elements in the dataset. (see section Conclusions in Davide
> Chicco and Giuseppe Jurman [1]).
>
> This is exactly what we need in our context.
>
> Q3: It is not perfectly clear how the annotations of the French Guiana forest have been built. Do you
> have any additional comment on this topic?
>
> Annotation was done on high-resolution Terrestrial Laser Scanning point cloud acquired simultaneously. The labels were subsequently transferred to the less dense ULS point cloud. The procedure used to annotate the TLS point cloud follows the description given in Martin-Ducup et al.[ 3] under their section entitled "Human assisted generation of control data".
>
> Response to weakness:
>
> (1) There is a lack of related works ...
>
> The Table 6 in Choe et al.[4 ] give such pertinent information. PointNet++ exhibits advantages in
> terms of memory consumption and training speed, with the results being relatively comparable. While
> I am inclined to conduct further experiments about the GPU consumption and the performances of
> different architecture, but resource constraints currently limit our capacity to do so.
>
> In essence, transformer-based architecture will do a lot of dot products to compute a sort of similarity
> score between the query and key vectors. This nature inherently leads to slower processing speeds.
>
> (2) An ablation study of the proposed geometric features ...
>
> We can share the results of an experiment which compares the full model with downgraded model
> versions where only a single geometric feature was used at a single scale (see Figure 4 and Table 1 in rebuttal_figures.pdf). Those results clearly show the benefits of including precomputed multiple geometric features at multiple scales. We think linearity and verticality contribute more to trunk recognition, whereas PCA1 and Sphericity are more efficient inside canopy.
>
> PointNet++ is able to learn more sophisticated features, but for empirically chosen geometric attributes, it may be not capable to learn identical ones.
>
> (3) ...using the reflectance as an additional feature per point.
>
> Indeed, reflectance may in some circumstances help discriminate leaf from wood and was used for instance in Wu et al.[5]. However, this makes the model sensor specific as different wavelengths are commonly used in different sensors.
>
> The main reason we decided not to use reflectance-derived information is its high variability and
> lack of specificity. In fact, reflectance cannot discriminate leaf from wood reliably at 905nm the
> wavelength at which our sensor operated (see for instance Figure 2 in Brede et al. [6]). In addition,
> multiple returns (pulse fragmentation), unknown object orientation, and surface wetness all contribute
> to adding noise to the apparent reflectance associated with each return(for a discussion of that point
> please see for instance Vincent et al. [7]).
>
> (4) The lack of experiments with competing methods...
>
> While we only compare our method to two others, the latter are commonly recognized as the best
> alternatives in the context we are dealing with (semantic segmentation of forest point clouds). A
> comparison extended to other DL methods with different architectures, which have no proven record
> of performance for the targeted task would be feasible, but we just did not have the resource to try
> that.
>
> Response to comments:
>
> We will correct all the typos accordingly in the revised version, and thanks for the practice, we will use this common practice to estimate sd in the future.
>
> References
>
> [1] D. Chicco and G. Jurman, “The advantages of the matthews correlation coefficient (mcc) over f1
> score and accuracy in binary classification evaluation,” BMC Genomics, vol. 21, 01 2020.
>
> [2] J. Yao and M. Shepperd, “Assessing software defection prediction performance,” in Proceedings
> of the Evaluation and Assessment in Software Engineering, ACM, apr 2020.
>
> [3] O. Martin-Ducup, I. Mofack, Gislain, D. Wang, P. Raumonen, P. Ploton, B. Sonké, N. Barbier,
> P. Couteron, and R. Pélissier, “Evaluation of automated pipelines for tree and plot metric
> estimation from tls data in tropical forest areas,” Annals of Botany, vol. 128, pp. 753–766, 04
> 2021.
>
> [4] J. Choe, C. Park, F. Rameau, J. Park, and I. S. Kweon, “Pointmixer: Mlp-mixer for point cloud
> understanding,” 2022.
>
> [5] B. Wu, G. Zheng, and Y. Chen, “An improved convolution neural network-based model for
> classifying foliage and woody components from terrestrial laser scanning data,” Remote Sensing,
> vol. 12, no. 6, 2020.
>
> [6] B. Brede, H. M. Bartholomeus, N. Barbier, F. Pimont, G. Vincent, and M. Herold, “Peering
> through the thicket: Effects of uav lidar scanner settings and flight planning on canopy volume
> discovery,” International Journal of Applied Earth Observation and Geoinformation, vol. 114,
> p. 103056, 2022.
>
> [7] G. Vincent, P. Verley, B. Brede, G. Delaitre, E. Maurent, J. Ball, I. Clocher, and N. Barbier,
> “Multi-sensor airborne lidar requires intercalibration for consistent estimation of light attenuation
> and plant area density,” Remote Sensing of Environment, vol. 286, p. 113442, 2023.

---

> > ### Comment · Reviewer_bCLj · 2023-08-12
> >
> > I would like to thanks the authors for their valuable rebuttal.
> >
> > Q2: thanks for the clarification, this should be included in the submission instead of "very good".
> >
> > W2: this ablation study is appreciated and should be integrated in the main submission document. It's an interesting to show that PointNet++ is no able to learn these features and could open up research in the application of more expressive backbones with comparable computational costs.
> >
> > General comment: since the annotated data will be release, the main submission document should identify it clearly as a contribution.
> >
> > Considering the new experiments and materials provided, I will increase my rating towards the acceptance.
> > Note that this paper provides methodological and practical contributions for forest monitoring while including a unique annotated dataset to motivate research in this field.
> >
> > I would like to highlight once again a comment provided in my review in Strengths, 1/: "The application of semantic segmentation to LiDAR recordings of forests is a high priority for climate change and global warming understanding and mitigation. This original work could have a potentially high impact for reforestation/afforestation monitoring and carbon stock estimation."

---

> > > ### Author Response · Authors · 2023-08-13
> > >
> > > Thank you for your positive feedback, we are happy that you find our response useful. We will incorporate the content you mentioned in the revised version.

---

### Official Review · Reviewer_yADz · 2023-07-04

**Soundness:** 3 good
**Presentation:** 3 good
**Contribution:** 3 good
**Rating:** 7
**Confidence:** 3

**Summary:**

This paper proposes a dataset and an algorithm for 3D semantic segmentation in forest scenes. From data collection to the algorithm design, this paper covers the whole pipeline that are oriented for forest segmentation. In terms of the algorithmic part, the solution itself is not fully satisfied with me, but I enjoyed the geometric feature computation (Sec.3.1). Moreover in data pre-partitioning part, I understand the intention of such design and it makes sense to me.

**Strengths:**

This paper covers the whole pipeline, from data collection to algorithm inference (semantic segmentation), for tree segmentation. While most of the 3D semantic segmentation methods, PointNet++, Point Transformer (ICCV 21), PointMixer (ECCV 22), PointNeXT (Neurips 2022) only deal with 3D semantic segmentation within the limited indoor scenes (S3DIS dataset or ScanNet dataset), this paper newly introduce the tree/forest segmentation using LiDAR point clouds.

Not just an algorithm part, this paper also covers data collection and data preprocessing such that this paper covers the whole pipeline for the forest segmentation task.

**Weaknesses:**

Honestly, I could not find the weakness of this paper. Nonetheless, I have a minor question. Can you conduct an ablation study for the rebalance loss? If possible, I want to see the quantitative/qualitative result based on this loss design.

Except that question, I am fully okay with this paper.



**Questions:**

Please check the weakness section above. Though I wrote down not much contents on this review, I think that this paper is quite solid and reasonable. I really enjoyed reading this submission.

**Limitations:**

It's fine with me.

---

> ### Author Rebuttal · Authors · 2023-08-08
>
> Thank you for your very positive and supportive review. We truly appreciate your acknowledgment of our efforts.
>
> Q1: Can you conduct an ablation study for the rebalance loss? If possible, I want to see the quantitative/qualitative result based on this loss design.
>
> Since the class imbalance problem is vital, we do early replace the cross entropy by focal loss, and we are certain that the cross entropy version will generally predict all the points as leaf. You can see the qualitative result of the focal loss version by referring to Figure 3 in rebuttal_figures.pdf. The qualitative results can be found in Table 1 of the rebuttal_figures.pdf.

---

> ### Comment · Reviewer_yADz · 2023-08-17
> **Post-rebuttal evaluation.**
>
> Thank the authors for the rebuttals. I also read the whole reviews and the corresponding rebuttals. I still think that this paper has its own contribution. While the reviewers, __eT4d__ and __uJjq__, tackle the (technical) novelty of this paper, I believe that the strength of this paper is more dependent on its unique problem setup, as the authors claim in the rebuttal, _Our paper is in line with "Machine learning for sciences (e.g. climate, health, life sciences, physics, social sciences)" of NeurIPS inviting topics, and we believe our work aligns well with the conference's interdisciplinary focus._.
>
> In my opinion, I understand the worrying points addressed by the reviewers who are against this paper. I also have experience in this field so the technical details and the proposed loss need to be carefully examined as an ablation study using the widely-known datasets, such as S3DIS. However, since this paper typically focuses on the 3D semantic segmentation in forest scenes, I believe that such strict analysis looks redundant to judge the novelty of this paper.
>
> __Let me vote for acceptance__. I really enjoyed reading this paper.

---

> > ### Author Response · Authors · 2023-08-17
> >
> > Thank you very much for your additional comment. We are very grateful for your support and enthusiastic about your analysis.

---

### Official Review · Reviewer_81in · 2023-07-05

**Soundness:** 3 good
**Presentation:** 2 fair
**Contribution:** 2 fair
**Rating:** 3
**Confidence:** 4

**Summary:**

They describe an approach for automatically segmenting a LIDAR scan of a forest into wood and leaf points.  They train a PointNet++ model and use resampling to address the extreme class imbalance in the data.  In comparison to previous methods they achieve a much higher balanced accuracy on their dataset.

**Strengths:**

The approach they propose is novel to the best of my knowledge.  The presentation is fairly detailed and clear.

According to their experimental results (Table 1), they greatly outperform previous methods in terms of specificity and balanced accuracy.

**Weaknesses:**

There is probably not enough of a contribution here in terms of machine learning methods for this paper to be appropriate for NeurIPS.  The paper is rather narrow in scope and specific to the application of wood-leaf segmentation.  Furthermore, they combine existing techniques such as extraction of PCA features, PointNet++, and resampling to handle imbalanced data.  As an application study, I would see this type of work as more appropriate for an applied machine learning conference / journal or a forestry / ecology journal.

Also, PointNet++ has been used before for wood-leaf segmentation -- this paper was not cited:

Xi, Zhouxin, et al. "See the forest and the trees: Effective machine and deep learning algorithms for wood filtering and tree species classification from terrestrial laser scanning." ISPRS Journal of Photogrammetry and Remote Sensing 168 (2020): 1-16.

Comments on presentation:
* L149 need spaces in the vector notation [0 0 1] (you can do $[0 ~ 0 ~ 1]$ for example)
* Eq. 7: if the voxels are adjacent, wouldn't the manhattan distance be 1 anyway?  How is "adjacent" defined?
* Figure 3: the numbers in the color bar are too small to read
* Eq. 9: the loss doesn't seem to be properly defined.  When y_k = 0, the loss term = 0?
* L254: "big trunk" -> "large trunks"
* Figure 5, 6: what do the colors mean here?  The statement "any other approaches developed for dense point cloud are ineffective" is too broad.

**Questions:**

It was not clear to me why they needed to first cluster the tree into large segments; perhaps they could better explain the motivation for this.

**Limitations:**

Some limitations were discussed but there was not a section specifically labeled "limitations."  Ethical implications were not discussed.

---

> ### Author Rebuttal · Authors · 2023-08-03
>
> Thank you for very helpful comments, and thank you for bringing this recent work[1] to our attention.
> We would add the discussion and cite the paper in the final revision.
>
> Q1: It was not clear to me why they needed to first cluster the tree into large segments; perhaps they
> could better explain the motivation for this.
>
> The ULS data often covers several hectares, or even hundreds of hectares. The situation in tropical
> forests is also highly complex, with various types and sizes of trees densely packed together. The
> significant memory demands make it nearly impossible to process all the data at once, leading us to
> adopt a spatial split schema approach as a necessary compromise.
>
> We can select data randomly from the whole scene, but we agree that selecting data randomly can
> result in a sparse and information-poor sample. Or we can employ a divide and conquer strategy to
> handle the chaotic, big volume, and complex ULS data. That’s why we propose GVD, a method that
> involves breaking down the data into more manageable subsets (refer to Figure 1 in rebuttal_figures.pdf), allowing us to handle the intricacies
> and extract meaningful insights in a more systematic manner. This approach enables us to retain the
> information-rich aspects of the data while overcoming computational challenges associated with the
> sheer volume of information.
>
> In our global response to all the reviewers, we have introduced an alternative spatial split scheme
> called the "sliding window". This method involves dividing the entire scene into multiple overlapping
> cuboids. Within each cuboid, we select training, validation, and testing samples. While this isn’t a
> strict ablation study of GVD, but this allows us to observe the clear impact of border effects (refer to Figure 2 in rebuttal_figures.pdf) and the disruption of spatial information in point cloud data.
>
> Response to weakness & comments:
>
> (1) weakness : There is probably not enough of a contribution here in terms of machine learning methods for this paper to be appropriate for NeurIPS. The paper is rather narrow in scope and specific to the application of wood-leaf segmentation.
>
> The proposed rebalanced loss addresses the persistent challenge of class imbalance in real-world
> data, which has practical implications in various domains. By presenting at NeurIPS, we aim to raise
> visibility and engage with experts in the field. Additionally, our GVD preprocessing method offers a
> new perspective for handling spatial dependencies. We firmly believe our contributions will benefit the research community, with potential applications beyond our immediate focus. Our paper is in line with "Machine learning for sciences (e.g. climate, health, life sciences, physics, social sciences)" of NeurIPS inviting topics, and we believe our work aligns well with the conference's interdisciplinary focus.
>
> Plus, as you mentioned, PointNet++ has been employed not only in the study[1] but also by
> others[2]. However, none of these works have successfully applied this architecture to ULS data.
> Furthermore, we have undertaken targeted adjustments to the PointNet++ backbone for our specific
> context (see section B, Architecture of DL model, in our paper).
>
> (2) Eq. 7: if the voxels are adjacent, wouldn’t the Manhattan distance be 1 anyway? How is "adjacent"
> defined?
>
> Adjacent voxels are defined as voxels that share a common surface, if the voxel a and voxel b are adjacent, their Manhattan distance is 1. Consider voxel b and voxel c are adjacent, while voxel a and voxel c are not. In this case, the Manhattan distance  (or geodesic distance in paper) between voxel a and voxel c is 2. What we want to calculate is the Manhattan distance between one of the lowest voxel, this voxel is fixed once it is chosen, and all the other voxels within the same component given by GVD. The voxel situated at the lowest position along the z-axis is called "lowest voxel."
>
> (3) Eq. 9: the loss doesn’t seem to be properly defined. When yk = 0, the loss term = 0?
>
> We will define it in a more proper way in the revised version.
>
> (4) Figure 5, 6: what do the colors mean here? The statement "any other approaches developed for
> dense point cloud are ineffective" is too broad.
>
> Red indicates high likelihood of being a wooden point, while blue indicates high likelihood of being
> a leaf point. The color gradient can be referred to in Figure 6’s color bar.
>
> Indeed, we will revise the statement in the final version.
>
> (5) typos problem
>
> We will correct all the typos accordingly in the revised version.
>
> References
>
> [1] Z. Xi, C. Hopkinson, S. B. Rood, and D. R. Peddle, “See the forest and the trees: Effective
> machine and deep learning algorithms for wood filtering and tree species classification from
> terrestrial laser scanning,” ISPRS Journal of Photogrammetry and Remote Sensing, vol. 168,
> pp. 1–16, 2020.
>
> [2] S. Krisanski, M. S. Taskhiri, S. Gonzalez Aracil, D. Herries, A. Muneri, M. B. Gurung, J. Mont-
> gomery, and P. Turner, “Forest structural complexity tool—an open source, fully-automated tool for measuring forest point clouds,” Remote Sensing, vol. 13, no. 22, 2021.

---

> > ### Comment · Reviewer_81in · 2023-08-16
> >
> > I have read over the other reviews and the authors' responses.  I would like to maintain my rating as I still think the level of contribution is below the bar for NeurIPS.

---

> > > ### Author Response · Authors · 2023-08-17
> > >
> > > Thanks for your additional feedback. We are sorry to read that you do not share our enthusiasm for this important topic and strongly believe that it would be important to attract more attention from the ML community to climate change related applications. In addition, we feel, we have provided response to your scientific and technical concerns. Are there any left issues that justify your decision to reject the paper?

---

> > > > ### Comment · Reviewer_81in · 2023-08-17
> > > >
> > > > I apologize for not being more specific.  I agree that work related to ML for science and climate change should be promoted at NeurIPS in general.  However there needs to be a sufficient level of contribution.  Regarding the claimed contributions:
> > > > 1. The "rebalancing loss" amounts to resampling the data at each batch to achieve a 1:1 class ratio.  This is a standard technique and not a novel contribution.  It is implemented in standard packages such as the WeightedRandomSampler in PyTorch.  A related standard technique would be class weighting which was not discussed.
> > > > 2. The changes to PointNet++ seem to be minor tweaks like changing the dropout rate, optimizer, etc.
> > > > 3. If they want to claim that data pre-partitioning is an important new technique for sparse point cloud segmentation in general, then I think they would need to perform more experiments testing the idea on different applications and comparing with state-of-the-art.  Right now, the experimental section is only focused on their particular application and only compares against other methods designed for the exact same task.

---

> > > > > ### Author Response · Authors · 2023-08-18
> > > > >
> > > > > Thank you for your feedback and valuable comments, we really appreciate this opportunity to address your concerns.
> > > > >
> > > > > Q1: The "rebalancing loss" amounts to resampling the data at each batch to achieve a 1:1 class ratio. This is a standard technique
> > > > > and not a novel contribution. It is implemented in standard packages such as the WeightedRandomSampler in PyTorch. A related
> > > > > standard technique would be class weighting which was not discussed.
> > > > >
> > > > > 1.1/ Rebalanced loss and WeightedRandomSampler operate through distinct mechanisms
> > > > >
> > > > > WeightedRandomSampler samples elements from training dataset with given weights, which means the number of elements in each
> > > > > batch for loss calculation remains is fixed. We can’t apply this practice to point cloud semantic segmentation, as it cannot ensure that
> > > > > points within each batch retain meaningful local geometric information, thereby break the underlying inter-point dependencies.
> > > > >
> > > > > Rebalanced loss first determines the number of wood points in each batch and then selects an equal number of leaf points from the
> > > > > batch provided by the GVD method. In other words, the count of elements participating in loss calculation varies in each iteration.
> > > > > Please note that the spatial distribution of data is not homogeneous, the data distribution within each batch is also not homogeneous.
> > > > > In the local regions constrained by GVD, within a batch, fewer wood points lead to a reduced selection of leaf points for loss
> > > > > calculation, which aids in semantically segmenting sparsely populated wood/leaf points in real data. Conversely, more wood points
> > > > > enable the selection of a higher number of leaf points in the local area, facilitating semantic segmentation of regions similar to the
> > > > > batch in real data. Regardless of any non-uniformity, our model learns under balanced conditions.
> > > > >
> > > > > 1.2/ class weighting method doesn’t work
> > > > >
> > > > > Focal loss has two factors $\alpha$ and $\gamma$, if $\gamma$ equals 0, focal loss is the class weighting method. We have tested it and the
> > > > > model still be overwhelmed by leaf points. The best result of using focal loss (class weighing method worse than the best result of focal loss) is provided in the Table 1 of our paper.
> > > > >
> > > > > Q2: The changes to PointNet++ seem to be minor tweaks like changing the dropout rate, optimizer, etc.
> > > > >
> > > > > Except these changes, our rebalanced loss + GVD yield substantial improvements in results. Akin to Qian et al.[1].
> > > > >
> > > > > Q3: If they want to claim that data pre-partitioning is an important new technique for sparse point cloud segmentation in general, then
> > > > > I think they would need to perform more experiments testing the idea on different applications and comparing with state-of-the-art.
> > > > > Right now, the experimental section is only focused on their particular application and only compares against other methods designed
> > > > > for the exact same task.
> > > > >
> > > > > Currently, we are unable to provide the comprehensive testing and ablation study on the GVD method across other datasets as you
> > > > > require, and we agree that drawing conclusions based on such comprehensive ablation study would indeed lead to fewer disputes.
> > > > > But we still hope you can reconsider the significant performance improvement from using GVD method + focal loss for this task.
> > > > >
> > > > > On August 10th, we obtained the latest version of the SOUL model specifically designed for TLS data. Although currently unreleased,
> > > > > this TLS-adapted SOUL model has already outperformed all other models on our available French Guyana TLS test dataset. The
> > > > > only difference between this TLS version and the ULS version mentioned in the paper is the adjustment in batch size, which has
> > > > > been increased from 5000 to 20000. Given the substantial spatial distribution differences between TLS and ULS data, this result
> > > > > potentially lends support to the efficacy of the GVD + focal loss approach.
> > > > > References
> > > > >
> > > > > [1] G. Qian, Y. Li, H. Peng, J. Mai, H. A. A. K. Hammoud, M. Elhoseiny, and B. Ghanem, “Pointnext: Revisiting pointnet++ with
> > > > > improved training and scaling strategies,” NeurIPS 2022.

---

### Official Review · Reviewer_uJjq · 2023-07-06

**Soundness:** 3 good
**Presentation:** 3 good
**Contribution:** 3 good
**Rating:** 4
**Confidence:** 4

**Summary:**

This paper introduces a neural network model based on the Pointnet ++ architecture which makes use of point geometry only (excluding any spectral information). To cope with local data sparsity, it proposes a sampling scheme that aims to preserve local important geometric information. It also proposes a loss function adapted to the severe class imbalance. Experiments show that the proposed model outperforms state-of-the-art methods on UAV point clouds.

**Strengths:**

The paper applies the mature pointnet++ on lidar tree classification with some variations of the methodology. The results seems good and working on the low resolution UAV LIDAR point clouds.

**Weaknesses:**

I am not quite sure of the novelty part as most of the techniques seem mature technique.

**Questions:**

Probably need to go beyond the pointnet++ methods which is relatively outdated.

**Limitations:**

The innovation part seems lacking.

---

> ### Author Rebuttal · Authors · 2023-08-04
>
> Thank you for your comments and  advice. We appreciate the opportunity to address your concerns.
>
> Q1: Probably need to go beyond the pointnet++ methods, which is relatively outdated.
>
> In fact, the recent work by Qian et al.[1] at NeurIPS 2022 demonstrates PointNet++ backbone’s enduring relevance. They achieved substantial performance gains through training strategy adjustments without any architectural changes, which underscores the architecture’s continued efficacy. Through a small modification, they reinstated the performance of PointNet++ to a state-of-the-art level. This aligns with our approach, akin to the targeted adjustments in SOUL, suggesting the potential for significant performance improvements with subtle modifications.
>
> Another significant advantage worth highlighting is PointNet++ has smaller latency, fewer parameters, and lower memory consumption (see Table 6 in Choe et al.[ 2]). Latency significantly influences training speed, and due to the use of rebalanced loss, SOUL demands a big number of training epochs. Hence, PointNet++’s advantage of lower latency proves highly advantageous for our purposes.
>
> Indeed, a comparison extended to other DL methods with different architectures would be feasible, but we just did not have the resource to try that.
>
> (1) Limitations & Weaknesses: The innovation part seems lacking.
>
> We agree that our individual ingredients may be established techniques. However, combining them in a way resulting on satisfying results on such challenging data set, required a certain amount of expertise and design that resulted in a novel proposal. This is also confirmed by the absence of real competitors among methods addressing similar applications and datasets.
>
> As mentioned in the global response and in the response to reviewers yADz and bCLj, the innovative parts of our proposal lie in (1) a whole pipeline for the forest segmentation task with potential community impact, (2) GVD preprocessing for sparse forest data and (3) rebalanced loss function designed for unbalanced ULS forest recordings. We will try to better emphasize these aspects in our revisions.
>
> References
>
> [1] G. Qian, Y. Li, H. Peng, J. Mai, H. A. A. K. Hammoud, M. Elhoseiny, and B. Ghanem, “Pointnext: Revisiting pointnet++ with improved training and scaling strategies,” 2022.
>
> [2] J. Choe, C. Park, F. Rameau, J. Park, and I. S. Kweon, “Pointmixer: Mlp-mixer for point cloud understanding,” 2022.

---

### Official Review · Reviewer_eT4d · 2023-07-06

**Soundness:** 2 fair
**Presentation:** 2 fair
**Contribution:** 2 fair
**Rating:** 3
**Confidence:** 5

**Summary:**

This submission proposes an end-to-end approach, Semantic segmentation On ULS (SOUL), for leaf-wood semantic segmentation that is based on PointNet++ [8]. By considering the imbalanced class label in the collected ULS dataset, a rebalanced loss is used. Moreover, a geodesic voxelization decomposition (GVD) method is introduced for data refinement through pre-partition. A ULS dataset with 282 tree-labels is collected for network training and testing. Experiments on the collected dataset demonstrate the effectiveness of the proposed method compared with the chosen baselines.

**Strengths:**

This submission proposes an end-to-end approach, Semantic segmentation On ULS (SOUL), for leaf-wood semantic segmentation that is based on PointNet++ [8]. By considering the imbalanced class label in the collected ULS dataset, a rebalanced loss is used. Moreover, a geodesic voxelization decomposition (GVD) method is introduced for data refinement through pre-partition. A ULS dataset with 282 tree-labels is collected for network training and testing. Experiments on the collected dataset demonstrate the effectiveness of the proposed method compared with the chosen baselines.

The strengths are:
1) A new dataset has been collected.
2) A new approach with comparable results.


**Weaknesses:**

The weaknesses of this paper are listed as follows:

1) The writing and the organization of the submission need to be improved.

2) The benefits of using the data pre-partitioning (in Section 3.2) are not clear. It would be better to provide more details and an ablation study w/o the GVD method.

3) The details of the provided baselines are missing. It would be better to consider more baselines in Table 1, e.g., PointNet++ with the proposed sub-modules.

4) The novelty is not sufficient for NeurIPS standards.

5) There are lots of approaches for imbalanced data labels. It would be better to provide more experiments to demonstrate the effectiveness of the rebalanced loss used.


**Questions:**

See Weaknesses.

**Limitations:**

See Weaknesses.

---

> ### Author Rebuttal · Authors · 2023-08-03
>
> Thank you for your valuable and constructive comments, we appreciate this opportunity to respond to your comments and address your concerns.
>
> Q1: The writing and the organization of the submission need to be improved.
>
> We will improve both in the revised version and check carefully and correct all typos.
>
> Q2: The benefits of using the data pre-partitioning (in Section 3.2) are not clear. It would be better to provide more details and an ablation study w/o the GVD method.
>
> Further elaboration on the benefits of the GVD method is warranted in our paper. ULS data often covers several hectares, or even hundreds of hectares. The situation in tropical forests is also highly complex, with various types and sizes of trees densely packed together. The significant memory demands make it nearly impossible to process all the data at once, leading us to adopt a spatial split scheme approach as a necessary compromise.
>
> We can select data randomly from the whole scene, but selecting data randomly can result in a sparse and information-poor sample. An alternative is to employ a divide and conquer strategy to handle the chaotic, big volume, and complex ULS data. That's why we propose GVD, a method that involves breaking down the data into more manageable subsets (refer to Figure 1 in rebuttal_figures.pdf), allowing us to handle the intricacies and extract meaningful insights in a more systematic manner. This approach enables us to retain the information-rich aspects of the data while overcoming computational challenges associated with the sheer volume of data.
>
> In our global response to all the reviewers, we have introduced an alternative spatial split scheme called the "sliding window". This method involves dividing the entire scene into multiple overlapping cuboids. Within each cuboid, we select training, validation, and testing samples. While this isn't a strict ablation study of GVD, this allows us to observe the clear impact of border effects and the disruption of spatial information in point cloud data (refer to Figure 2 in rebuttal_figures.pdf). This demonstrates the advantages of the GVD method.
>
> Q3: The details of the provided baselines are missing. It would be better to consider more baselines in Table 1, e.g., PointNet++ with the proposed sub-modules.
>
> In fact, SOUL is the first approach to tackle semantic segmentation of tropical forest ULS data. SOUL might serve as the first baseline in this field. We have introduced both the best-performing unsupervised method (LeWoS) and a deep learning approach (FSCT) tailored for TLS data. The latter are commonly recognized as the best alternatives in the context we are dealing with (semantic segmentation of forest point clouds). These methods do not perform well on ULS data, mainly because they were not specifically designed to handle the raw ULS data and address class imbalance issues. Our qualitative evaluation has demonstrated that our methods represent the state-of-the-art in terms of performance.
>
> It is worth noting that access to code is limited in this domain, hindering reproducibility and comparative analysis. Moreover, certain methods rely on intensity data, which can vary based on different devices. Consequently, the generalizability of testing results remains uncertain. That's the reason why we aim at using solely coordinates as input of SOUL. To foster collaboration and facilitate meaningful exchange within the research community, we have made the decision to openly share our code and data, hoping to engage with fellow researchers in a more inclusive and productive manner.
>
> Q4: The novelty is not sufficient for NeurIPS standards.
>
> We appreciate the reviewer's feedback and understand their concern regarding the novelty of our work. Nonetheless, we would like to highlight that our research does indeed bring valuable contributions to the field. The proposed method GVD is also relevant for other LiDAR sensors and type of forests and the rebalanced loss addresses the challenge of class imbalance in real-world data, which may have practical implications in various domains.
>
> As mentioned in the global response and in the response to reviewers yADz and bCLj, the innovative parts of our proposal lie in (1) a whole pipeline for the forest segmentation task with potential community impact, (2) GVD preprocessing for sparse forest data and (3) rebalanced loss function designed for unbalanced ULS forest recordings. We will try to better emphasize these aspects in our revisions.
>
> Q5: There are lots of approaches for imbalanced data labels. It would be better to provide more experiments to demonstrate the effectiveness of the rebalanced loss used.
>
> We have provided a more comprehensive comparison between the focal loss and the rebalanced loss in our global response to all reviewers and in the final version. Additionally, we will publish our code and data, hoping for further interactions with researchers in the field.

---

> > ### Comment · Reviewer_eT4d · 2023-08-21
> >
> > Thank you for the authors' responses. We acknowledge the dataset, the main contribution of the submission and should be highlighted in their main paper, might have a potentially high impact on reforestation/afforestation monitoring. However, the technique novelty does, in my view, not meet the bar of Neurips as the method is more likely an approach that borrows existing ML techniques for a specific task. Unless the technical details can be compared with SOTA baselines on other public datasets and with a proper ablation study, I will maintain my rating.

---

> > > ### Author Response · Authors · 2023-08-21
> > >
> > > Thank you for your feedback. While we understand your concerns, we would like to emphasize the difference of our application context compared to existing well-known datasets. Unlike point cloud data of mock-up generated objects, indoor scenes, and autopilot datasets, large-scale raw forest point cloud data exhibits sparsity, irregularity, and heterogeneity. An experienced researcher in this domain cannot ensure accurate classification of wood and leaf points even in artificial way.
> > >
> > > Considering the results given by SOUL in comprehensive testing across various analogous datasets: UAV data (ULS), ground-based equipment data (TLS), and backpack data (MLS), we share reviewer yADz's viewpoint regarding the ablation study aspect: "...the technical details and the proposed loss need to be carefully examined as an ablation study using the widely-known datasets, such as S3DIS. However, since this paper typically focuses on the 3D semantic segmentation in forest scenes, I believe that such strict analysis looks redundant to judge the novelty of this paper."

---

### Author Rebuttal · Authors · 2023-08-07

We thank the reviewers for their comments and for underlining the existence of new and good
features in our work, in particular regarding new methodological contributions, reproducibility and
relevance of the experiments, while also suggesting improvement and clarification. We also note that
the importance of forests segmentation, as an essential step to climate change and global warming
understanding and mitigation, has been acknowledged. We summarize below the main points addressed in our response:

(1) GVD ablation study:

Further elaboration on the benefits of the GVD method is warranted in our paper. The ULS data often covers several hectares, or even hundreds of hectares. The situation in tropical forests is also highly complex, with various types and sizes of trees densely packed together. The significant memory demands make it nearly impossible to process all the data at once, leading us to adopt a spatial split scheme approach as a necessary compromise.

We can select data randomly from the whole scene, but selecting data randomly can result in a sparse and information-poor sample. An alternative is to employ a divide and conquer strategy to handle the chaotic, big volume, and complex ULS data. That's why we propose GVD, a method that involves breaking down the data into more manageable subsets (refer to Figure 1 in rebuttal_figures.pdf), allowing us to handle the intricacies and extract meaningful insights in a more systematic manner. This approach enables us to retain the information-rich aspects of the data while overcoming computational challenges associated with the sheer volume of data.

Prior to employing the GVD method, we initially adopted a more intuitive approach. The data was partitioned in unit of voxel, serving as component for batch preparation through down-sampling. However, this approach gives rise to border effects, particularly impeding SOUL's focus on the meticulous segmentation of intricate branch and leaf within tree canopy. The segmentation of cubes led to the emergence of noise point cluster along the voxel edges. ULS have more recording on tree canopy, so the presence of noise point cluster on voxel edges is more severe on tree canopy, which imposes bigger obstacles to achieving our goal.

We experimented with cuboids, a choice aimed at preserving greater semantic information within each component sample and expanding the spatial range for batch selection. Similar to a "sliding window", we can systematically traverse the entire forest with overlapping coverage in this way. But border effects persisted (see Figure 2 in rebuttal\_figures.pdf), prompting the introduction of the GVD method, which led to a substantial improvement. A comparison between the result of the downgraded version using sliding window as spatial split schema and the individual-component point performance of full version SOUL is illustrated (see Figure 1 in rebuttal\_figures.pdf). Notably, in component N°552, SOUL effectively discriminate trunk points from leaf points that were previously entirely intertwined, surpassing our initial expectations. So, in alignment with the observations of Reviewer bCLj, we hold the conviction that the proposed GVD preprocessing method is relevant beyond the confines of the present study, extending to various LiDAR sensors and diverse forest types.

A new video (3D version of Figure 2 in rebuttal_figures.pdf) demonstrating the use of cuboid as a spatial split method has been provided to AC.

(2) Novelty:

Our main contribution lies in the complete handling of the full pipeline from data
collection to final segmentation. To our knowledge, this is quite unique in forestry applications,
where either datasets are limited to much simpler structures, or analysis techniques show limited
performance. In this work, we combine advanced and original deep learning methods and principles
to get what is a first baseline on segmentation of ULS LiDAR data. Plus, our model SOUL exclusively depends on coordinates and does not incorporate RGB information, which is different from indoor scenes or autopilot contexts.

(3) Relevance to NeurIPS:

Our work makes use and proposes new developments and contributions in
core deep learning techniques. Although it focuses on point cloud processing, we believe that some
of the principles and ideas are transferable to other data structures and problems. As regards the
target application, according to this year call for papers, the conference scope includes explicitly life
and natural sciences. In addition, in contrast to some of the reviewers, we believe the topic of our
work may be of interest to the NeurIPS readership, especially as mentioned above, for its potential high
environmental applications impact, which is a rising concern for a lot of junior and senior researchers.
In addition, to foster collaboration and facilitate meaningful exchange within the research community,
we have made the decision to openly share our code and data, hoping to engage with fellow researchers
in a more inclusive and productive manner on these important applications.

(4) The video provided before:

Just in case, a video hyperlink has been embedded within the closing word of the first paragraph in the supplementary. This video shows the qualitative result of SOUL model and some comparisons between different methods.

---

### Decision · Program_Chairs · 2023-09-21

**Decision:**

Accept (poster)

**Comment:**

This paper introduces a dataset and methods for segmentation of sparse point clouds from UAVs for monitoring vegetation. The problem is urgently relevant and a good match for machine learning, and poses unique challenges. The biggest concern raised was in "novelty", but I concur with several of the reviewers in seeing this paper as an effective example of application-guided innovation, where the methodological advances were appropriately motivated by application-specific challenges and criteria for success. Given the utility of the dataset to the machine learning community and the solid baseline methods, I recommend the paper for acceptance. However, I do believe that adding as many comparison methods and ablations as possible in the camera-ready will improve the usefulness to the NeurIPS community.